# Modelling black carbon absorption of solar radiation: combining external and internal mixing assumptions

Gabriele Curci[1,2], Ummugulsum Alyuz[3], Rocio Barò[4], Roberto Bianconi[5], Johannes Bieser[6], Jesper H. Christensen[7], Augustin Colette[8], Aidan Farrow[9], Xavier Francis[9], Pedro Jiménez-Guerrero[4], Ulas Im[7], Peng Liu[10], Astrid Manders[11], Laura Palacios-Peña[4], Marje Prank[12,13], Luca Pozzoli[3,13], Ranjeet Sokhi[9], Efisio Solazzo[14], Paolo Tuccella[1,2], Alper Unal[3], Marta G. Vivanco[15], Christian Hogrefe[16], Stefano Galmarini[14]

[1] Department of Physical and Chemical Sciences, University of L'Aquila, L'Aquila, Italy
[2] Center of Excellence in Telesening of Environment and Model Prediction of Severe Events (CETEMPS), University of L'Aquila, L'Aquila (AQ), Italy
[3] Eurasia Institute of Earth Sciences, Istanbul Technical University, 34469 Istanbul, Turkey
[4] Department of Physics, University of Murcia, Murcia, 30003, Spain
[5] Enviroware s.r.l., Concorezzo (MB), 20863, Italy
[6] Helmholtz-Zentrum Geesthacht, Zentrum für Material- und Küstenforschung GmbH, Geesthacht, 21502, Germany
[7] Atmospheric Modelling Secton (ATMO), Department of Environmental Science, Aarhus University, Frederiksborgvej 399, 4000 Roskilde, Denmark
[8] Atmospheric Modelling and Environmental Mapping Unit, INERIS, BP2, Verneuil-en-Halatte, 60550, France
[9] Centre for Atmospheric and Instrumentation Research (CAIR), University of Hertfordshire College Lane, Hatfield, AL10 9AB, United Kingdom
[10] NRC Research Associate at Computational Exposure Division, National Exposure Research Laboratory, U.S. Environmental Protection Agency (EPA), Research Triangle Park, NC 27711, United States of America
[11] TNO, PO Box 80015, 3508 TA Utrecht, The Netherlands
[12] Finnish Meteorological Institute, Atmospheric Composition Research Unit, Helsinki, 00560, Finland
[13] Cornell University, Department of Earth and Atmospheric Sciences, Ithaca, 14853, NY, United States of America
[14] Joint Research Centre (JRC), European Commission, Ispra (VA), 21027, Italy
[15] CIEMAT, Madrid, 28040, Spain
[16] Computational Exposure Division, National Exposure Research Laboratory, U.S. Environmental Protection Agency (EPA), Research Triangle Park, NC 27711, United States of America

*Correspondence to*: Gabriele Curci (gabriele.curci@aquila.infn.it)

**Abstract.** An accurate simulation of the absorption properties is key for assessing the radiative effects of aerosol on meteorology and climate. The representation of how chemical species are mixed inside the particles (the mixing state) is one of the major uncertainty factors in the assessment of these effects. Here we compare aerosol optical properties simulations over Europe and North America, coordinated in the framework of the third phase of the Air Quality Model Evaluation International Initiative (AQMEII), to one year of AERONET sunphotometer retrievals, in an attempt to identify a mixing state representation that better reproduces the observed single scattering albedo and its spectral variation. We use a single post-processing tool (FlexAOD) to derive aerosol optical properties from simulated aerosol speciation profiles, and focus on the

absorption enhancement of black carbon when it is internally mixed with more scattering material, discarding from the analysis scenes dominated by dust.

We found that the single scattering albedo at 440 nm ($\omega_{0,440}$) is on average overestimated (underestimated) by 3-5% when external (core-shell internal) mixing of particles is assumed, a bias comparable in magnitude with the typical variability of the quantity. The (unphysical) homogeneous internal mixing assumption underestimates $\omega_{0,440}$ by ~14%. The combination of external and core-shell configurations (partial internal mixing), parameterized using a simplified function of air mass aging, reduces the $\omega_{0,440}$ bias to -1/-3%. The black carbon absorption enhancement ($E_{abs}$) in core-shell with respect to the externally mixed state is in the range 1.8-2.5, which is above the currently most accepted upper limit of ~1.5. The partial internal mixing reduces $E_{abs}$ to values more consistent with this limit. However, the spectral dependence of the absorption is not well reproduced, and the absorption Angstrom exponent $AAE_{675}^{440}$ is overestimated by 70-120%. Further testing against more comprehensive campaign data, including a full characterization of the aerosol profile in terms of chemical speciation, mixing state, and related optical properties, would help in putting a better constraint on these calculations.

## 1 Introduction

Aerosols suspended in the atmosphere interact with solar and planetary radiation and with clouds, influencing the Earth's energy balance, and gaps in the understanding of these interactions continue to contribute some of the largest uncertainties in projected climate change (Boucher et al., 2013). One important detail is how the different chemical species are spatially arranged inside each particle or, in other words, the knowledge of their mixing state (Fierce et al., 2017). Here we use an ensemble of regional model simulations over Europe and North America to compute aerosol optical properties under different mixing state assumptions and compare the resulting absorption properties with ground-based sunphotometer observations, in order to assess the most likely mixing state, or combination of mixing states.

In addition to changing the path of radiation from the incident beam (scattering), some aerosols may capture energy from the impinging radiation (absorption) and release it as thermal radiation. The resulting change of the radiative flux is called "radiative effect due to aerosol–radiation interactions (*REari*)" (formerly known as "direct radiative effect", Boucher et al., 2013). The heating of air due to the release of the absorbed energy is called "semi-direct effect", because it is linked to the local alteration of the atmosphere' static stability and cloud cover (Koch and Del Genio, 2010; Wilcox, 2010). Aerosols also serve as necessary condensation nuclei for cloud droplets and ice crystals to form in Earth's atmosphere. The overall impact on radiative fluxes, due to the change in cloud processes consequent to a change in aerosol concentrations, is called "'effective radiative forcing due to aerosol–cloud interactions (*ERFaci*)" (formerly known as "indirect radiative effect", Boucher et al., 2013). Moreover, aerosols may darken snow and ice surfaces, accelerating the melting rate through enhanced absorption of solar radiation (Pitari et al., 2015).

Mixing state of particles is key to an accurate estimate of both *REari* and *ERFaci* (Fierce et al., 2017). The mixing state describes how different chemical components are blended together and arranged in a single aerosol particle. At one extreme,

all compounds are separated, and each particle is made only of one species or aerosol type (external mixing). At the other extreme, all compounds are perfectly stirred in internal homogeneous mixing. In the real atmosphere, particles are expected to be somewhere in between these two extremes. The related uncertainty on calculated optical properties, such as aerosol optical depth and single scattering albedo, is of the order of 30-35% on a monthly mean basis (Curci et al., 2015). The mixing state of

5 black carbon (BC) is of particular relevance, because it has a large absorption power (mass absorption cross section of at least 5 $m^2$ $g^{-1}$ at a wavelength of 550 nm, Bond et al., 2013), which may be further amplified when it is coated with less absorbing material through a kind of "lensing effect" (Lesins et al., 2002). Observations of mixing state in the global atmosphere have been carried out in recent years by means of single particle aerosol mass spectrometry. Black carbon is usually found to be externally mixed within a few hours from emission, while internal mixing with sulfate-ammonium-nitrate and organic carbon

is very common in aged aerosol (Cheng et al., 2006; Pratt and Prather, 2010; Bi et al., 2011; Ma et al., 2017; Muller et al., 2017).

One popular and physically reasonable way to represent BC internal mixing is the core-shell model, where a BC core is surrounded by a shell of soluble material, such as sulfate or organic carbon. Early work suggested a global average BC absorption enhancement factor ($E_{abs}$) of ~2 (Jacobson, 2001), while subsequent studies at the regional scale reported a wide

range of $E_{abs}$ values, from negligible (~1, Cappa et al., 2012) to as high as ~2.4 (Peng et al., 2016), with many in the range of 1.2-1.6 (Bond et al., 2006; Schwartz et al., 2008; Moffet and Prather, 2009; Liu et al., 2017b). Current estimates of *REari* attributable to BC are on the order of 0.9 W $m^{-2}$, second only to *REari* of $CO_2$, but the uncertainty associated with $E_{abs}$ yields a poorly constrained range of 0.2-1 W $m^{-2}$ (Gustafsson and Ramanathan, 2016).

The other important aspect is the ability of aerosols to act as cloud condensation nuclei (CCN), which depends on their ability

to take up moisture from air (hygroscopicity). When mixtures of several components are present, the resulting hygroscopicity parameter of particles κ (Petters and Kreidenweis, 2007) depends on their mixing state. If an internal homogeneous mixture is assumed, the hygroscopicity is taken as the largest of all components, and κ (and so the number of CCN) may be greatly overestimated. On the other hand, if information on the fraction of less hygroscopic material is included, calculations are much more accurate (Wex et al., 2010). Knowledge on the mixing state is more important in the transition from fresh to aged aerosol.

The difference between external and internal mixing becomes small when hydrophobic material 100 nm in diameter has a coating of soluble material of 3 nm or more, which may be achieved in a few hours in photo-chemically active environments, such as urban areas during daytime (Wang et al., 2010). The total *ERFaci* is currently estimated as -0.45 (-1.2 to 0.0) W $m^{-2}$ (Myhre et al., 2013).

In this work, we use a suite of 11 regional scale air quality simulations over Europe and North America for the year 2010,

carried out in the framework of the third phase of the Air Quality Model Evaluation International Initiative (AQMEII, http://aqmeii.jrc.ec.europa.eu/, Galmarini et al., 2017), to compare calculated aerosol optical properties with observations from the sunphotometers' Aerosol Robotic Network (AERONET, https://aeronet.gsfc.nasa.gov/, Holben et al., 2001). As detailed in section 2, the aerosol optical calculations for the species profiles simulated by the individual regional scale air quality models use a single post-processing tool (FlexAOD, Curci et al., 2015, http://pumpkin.aquila.infn.it/flexaod/), in order to harmonize

the assumptions made in the optics calculations. Three basic physical quantities, commonly used in radiative transfer modelling, are derived and compared to column-wise sunphotometer observations: aerosol optical depth, single scattering albedo and asymmetry parameter. Special attention is devoted to absorption properties of aerosols, in particular those related to black carbon as a function of its mixing state. Two extreme cases are considered (external mixing and core-shell internal mixing), plus a combination of them weighted by a simple parameterization of aerosol aging (Cheng et al., 2012, see section 2). The comparison (section 3) focuses on the observed scenes where the influence of black carbon on absorption is estimated to be predominant. Finally (section 4), we discuss and summarize the observational constraints on the spatial-temporal distribution of the aerosol mixing state.

## 2 Data and methods

### 2.1 AERONET sunphotometer observations

In Figure 1 and Table 1, we show the location of the AERONET sunphotometers selected for the year 2010 over Europe and North America. We select only those stations having a minimum of 10% of valid data in 2010. Since our focus is on aerosol absorption properties, we use version 2 inversion products (Dubovik and King, 2000) which, in addition to the spectral (at nominal wavelengths $\lambda = 440, 675, 870, 1020$ nm) aerosol optical depth ($\tau(\lambda)$), provide estimates of the single scattering albedo ($\omega_0(\lambda)$) and the asymmetry parameter ($g(\lambda)$), among other quantities. The cloud-screened and quality-assured data are those labelled Level 2.0 (Dubovik et al., 2002), and we start from this dataset. Absorption retrievals for scenes having $\tau(\lambda=440$ nm) < 0.4 are automatically discarded in Level 2.0, because they are considered too uncertain (Dubovik et al., 2002). The uncertainty associated with the single scattering albedo is estimated to increase from ±0.03 for $\tau(\lambda=440$ nm) $\geq 0.5$ to ±0.05-0.07 for $\tau(\lambda=440$ nm) $\leq 0.2$ (Dubovik et al., 2000). The result is that more than 90% of absorption related observations in Level 2.0 data are discarded over regions with relatively low values of $\tau$: for year 2010, the median $\tau(\lambda=440$ nm) is 0.15 (0.08-0.24 interquartile range) over Europe and 0.08 (0.05-0.13) over North America. Similar to what was done by Wang et al. (2016), we thus add Level 1.5 absorption data to the dataset, so as to reinforce the model to observation comparison statistic. In Figures S1 and S2 we show the timeseries of $\tau$ and $\omega_0$ at 440 nm for each site.

Aerosol absorption in the visible-near infrared part of the solar spectrum is primarily determined by black carbon (BC), brown carbon (BrC), and mineral dust (Bergstrom et al., 2007). In this study, we attempt to impose an observational constraint on the simulated absorption due to black carbon, specifically in terms of the absorption enhancement attributable to its progressive internal mixing as it ages in the atmosphere. We thus select those AERONET scenes in which the contribution to the absorption by dust can be considered to be minimal, following the selection criteria suggested by Badhur et al. (2012). We define:

$$\tau_{sca}(\lambda) = \tau(\lambda) \, \omega_0(\lambda) \tag{1}$$

$$\tau_{abs}(\lambda) = \tau(\lambda) \, (1 - \omega_0(\lambda)) \tag{2}$$

$$SAE_{675}^{440} = -\frac{\ln\left(\frac{\tau_{sca}(\lambda=440)}{\tau_{sca}(\lambda=675)}\right)}{\ln\left(\frac{440}{675}\right)} \tag{3}$$

$$AAE_{675}^{440} = -\frac{\ln\left(\frac{\tau_{abs}(\lambda=440)}{\tau_{abs}(\lambda=675)}\right)}{\ln\left(\frac{440}{675}\right)} \tag{4}$$

where $\tau_{sca}$ and $\tau_{abs}$ are the scattering and absorption aerosol optical depths, and SAE and AAE are the scattering and absorption Ångström exponents, respectively. Following Badhur et al. (2012), scenes having a SAE $\leq$ 1.2 are labelled as "Dust"-dominated, those having SAE > 1.2 and AAE < 1.2 as "BC"-dominated, and the remaining as "BC+BrC"-dominated. The threshold on SAE, effectively separates coarse (dust-dominated) from fine (carbonaceous-dominated) absorbing particles (see Figures S3 and S4). We note that the method used here should be considered effective for segregating dust- and carbonaceous-dominated scenes, however recent work proposed improvements for a more quantitative segregation of the carbonaceous-dominated scenes in "BC" and "BrC" contribution (Wang et al., 2016).

In Figure 1 and Table 1, we display the relative fraction of the three absorption classes for each station. The monthly fractions at each site is shown in Figures S5 and S6. Over both continents the majority of observations are BC-dominated (>60%), and the vast majority of sites have a relative higher proportion of BC absorption class (16/20 in Europe, 7/9 in North America). This fact points out a dominant role of fossil fuel use in determining the absorption properties of aerosol over both continents. Two sites in Southern Spain (Huelva and Malaga) and three in North America (Egbert, El Segundo and Railroad Valley) have significant contribution from "Dust" scenes, because they are all subject to frequent advection from nearby arid areas (e.g. Sahara desert in Africa and Arizona deserts in the United States). Two sites in Europe (Barcelona and Munich) and one in North America (Egbert) have a prevalence of "BC+BrC" observations, possibly related to the significant impact of biomass burning, bio/solid fuel use, and secondary organic aerosol production.

## 2.2 AQMEII regional scale simulations

In Table 2 we list the main characteristics of the regional scale simulations carried out in the frame of the third phase of the Air Quality Model Evaluation International Initiative (AQMEII, http://aqmeii.jrc.ec.europa.eu/, Galmarini et al., 2017). Nine simulations are available over Europe and two over North America. Models share the same anthropogenic emission inventories, which were already used in phase 2 of AQMEII (Pouliot et al., 2015) and the same boundary conditions (BASE case in Galmarini et al., 2017) from the C-IFS model (Flemming et al., 2015). Some models use the sectional approach and others the modal approach to solve aerosol processes. Here, we use bulk concentrations (summed over all sizes and modes) to simulate optical properties, thus the difference may be relevant only when interpreting the diversity of simulated aerosol species profiles. As explained in the next section, optical calculations are carried out assuming the same size distributions (and other physical and chemical quantities) for all models. The native grid spacing and domain projections were specific of each model, but outputs were remapped onto a single grid for each continent at a horizontal resolution of $0.25° \times 0.25°$. Aerosol profiles on 18 layers (up to 9 km) were extracted for each hour of the year over AERONET locations and data delivered to a

common database (ENSEMBLE, http://ensemble.jrc.ec.europa.eu/) hosted by the Joint Research Centre (JRC) (e.g. Galmarini et al., 2012).

Most model simulations analysed in this study have been evaluated in terms of their skills in reproducing seasonal patterns of ground-level pollutants (Im et al., 2018), temporal and spatial patterns of ground- and upper-level concentrations (Solazzo et al., 2017), and wet and dry deposition processes (Vivanco et al., 2018). A general underestimation of surface total PM was found over both continents, particularly in winter. The underestimation is confirmed by Table S1, which shows the observed and modelled PM2.5 average values in 2010 at the available surface monitoring stations over Europe and North America in the ENSEMBLE database (~1000 stations for each continent). In Europe, the ES1 model is the only having average values slightly above the observations (due to a known overestimation of desert dust), all the others underestimate PM2.5 by 10 to 60%. In North America, the US3 model has almost no bias, while DK1 underestimates PM2.5 by 25%, mostly attributable to missing secondary organic aerosol mass. As explained in previous section, in the following we will focus our attention on scenes dominated by BC and BrC, thus discarding dust-dominated scenes. The comparison presented in Table S1 includes all the available scenes, since there is no straightforward way to separate BC, BrC, and dust contributions based on standard PM2.5 mass measurements, and thus must be taken just as a general guidance for the analysis of the simulated aerosol optical depth.

Additional indications about models skills are gathered from the comparison with PM composition measurements available near the AERONET stations, for which we have stored the simulated PM speciation profiles of AQMEII models. The comparison is carried out at 3 stations over Europe and 5 stations over North America, and results summarized in Table S1 and Figure S7. Over North America, the two models have yearly average values mostly within $\pm 1$ µg m$^{-3}$. Over Europe, most values are also within the same range, but there is a tendency toward overestimation of inorganic secondary species (sulfate, nitrate, ammonium) and black carbon, and underestimation of the organic carbonaceous fraction.

Figure 2 and Figure 3 show the model profiles averaged in space and time at AERONET observational sites for the year 2010. All models predict an exponential decay of aerosol species concentrations from the ground to the upper troposphere. Two models (FRES1 and NL1) have top height below 5 km, but above that altitude the aerosol concentrations are already generally low enough to make only a minor contribution to extinction in the troposphere. Most models simulate an average concentration of secondary inorganic species (sulfate, nitrate, ammonium) near the surface between 1 and 2 µg m$^{-3}$, with the exception of ES1 and TR1 which predict values around 4 µg m$^{-3}$. These two models are also those having the smallest difference against observed PM2.5 over Europe. Black carbon concentrations near the surface are mostly in the 0.2-0.6 µg m$^{-3}$ range, except for FI1 and TR1 that have values above 1 µg m$^{-3}$. Primary organic carbon concentrations are mostly around 1 µg m$^{-3}$, with models DE1 and UK3 below 0.5 µg m$^{-3}$ and models NL1 and TR1 above 1.5 µg m$^{-3}$. The secondary organic fraction displays the highest degree of model diversity, with most models simulating values below 0.2 µg m$^{-3}$, and IT2 and FI1 having average concentrations near the surface of about 1 and 5 µg m$^{-3}$, respectively. FI1 has also a relatively small bias of about 25% with respect to PM2.5 surface observations. Some models (DE1, DK1, NL1 and TR1) did not simulate secondary organic aerosol

or did not provide results for this component to the common database. The simulated values over North America are generally at the lower edge compared to those over Europe.

The figures also show the ratio rBC of the sum of secondary inorganics and total organics (primary plus secondary) to black carbon concentrations:

$$rBC = \frac{[SIA]+[OC]}{[BC]} \qquad (5)$$

Below 1 km, most models are in the range of 5-10, while above 1 km model dispersion increases. For most models, rBC increases monotonically with height up to values of 20-40, while for others (FI1 and IT2 over Europe, and DK1 over North America) it reaches a maximum in the free troposphere and then decreases upwards, possibly reflecting diversity in simulated aerosol aging and loss processes. The profiles of the calculated optical properties are discussed in more details in section 3, but here we anticipate that rBC is found to be proportional to the single scattering albedo and to the BC absorption enhancement ($E_{abs}$) mentioned in the introduction, while it is inversely proportional to the BC core mass fraction. Here we calculate the BC absorption enhancement as the ratio of absorption optical depth calculated assuming internal mixing to the one calculated using external mixing:

$$E_{abs} = \frac{\tau_{abs}(\lambda, internal\ mixing)}{\tau_{abs}(\lambda, external\ mixing)} \qquad (6)$$

The BC core mass fraction is defined for core-shell calculations as the ratio of BC mass (the core) to total aerosol mass (shell + core).

## 2.3 FlexAOD aerosol optical properties calculations

We use a single tool to derive aerosol optical properties from the aerosol chemical species mass profiles simulated by the various regional scale models. There are two main reasons for this choice: (1) most of the participating models have an internal algorithm to compute the aerosol optical depth, but, among those, not all also calculate the absorption properties (e.g. single scattering albedo); (2) the assumptions made for aerosol optical properties calculations differ among models, making any inter-comparison more difficult and ambiguous to interpret. The point is illustrated in Table S1 which shows the annual average values of PM2.5 and $\tau_{555}$ as calculated and reported by several of the regional scale models. The ES1 model has a PM2.5 average concentration very close to observations, but the aerosol optical depth is about twice that observed. The other four models for which the aerosol optical depth was available have different PM2.5 average values, but almost identical aerosol optical depths on the respective continent of application.

Therefore, we build on the methodology adopted in phase 2 of AQMEII, which employed the post-processing tool FlexAOD (Curci et al., 2015, http://pumpkin.aquila.infn.it/flexaod/) in order to apply a homogeneous set of assumptions to all models. We calculate aerosol optical properties assuming spherical particles and applying Mie theory (Mie, 1908). We assign to each chemical species considered a particle density, a dry complex refractive index, a hygroscopic growth factor, and a log-normal distribution. We list the parameters used to define the mentioned physical and chemical properties in Table 3 (source of data

are Highwood (2009) and Hess et al., (1998), the latter for hygroscopic growth factors), while the procedure to derive the aerosol optical depth, the single scattering albedo and the asymmetry parameter is detailed in Curci et al. (2015).

Specifically regarding the modelling of BC shape and mixing state, here we adopt the simplified approach widely used in regional and global models of assuming spherical particles and centred core-shell arrangement for internal mixing calculations, which makes the computation fast enough for 3-D applications in year-long simulations. However, observations show that BC in the real atmosphere displays a wide variety of shapes: freshly emitted hydrophobic fractal aggregates, consisting of hundreds of spherules having diameters of a few tens of nm (e.g. Posfai et al., 2003, Adachi and Buseck, 2013), typically evolve in the atmosphere assuming more compact structures, and internally or semi-internally coating with hydrophilic material (e.g. Adachi et al., 2010, China et al., 2015, Wang et al., 2017). These transformations affect the variability of the absorption properties of BC, as illustrated in several numerical studies that include detailed description of the shapes and mixing state of BC and that use advanced algorithms, such as the multiple-sphere T-matrix (MSTM) and the discrete dipole approximation (DDA), to compute the optical properties (Scarnato et al., 2013, He et al., 2015, He et al., 2016, Li et al., 2016, Kahnert, 2017, Liu et al., 2017a, Liu et al., 2018, Liu and Mishchenko, 2018). Moreover, also the shapes of BrC may vary in the real atmosphere, but their classification and investigation of numerical aspects in the calculation of optical properties is still at its beginning (Laskin et al., 2015, Liu and Mishchenko. 2018).

In Table 4 we list the sensitivity calculations we carried out to test the effect of mixing state on aerosol absorption properties. The reference case assumes external mixing of chemical species (EXT): in that case, the optical properties are calculated separately for the species listed in Table 3, and then summed or averaged, as detailed in Curci et al. (2015). For internal mixing cases, the volume average refractive index as a function of particle size must be computed before application of the Mie algorithm. The size range spanned by the log-normal distributions attributed to the species ($10^{-3}$ to 10 μm here) is divided into 100 geometrically spaced bins, and the mass of each aerosol species is calculated in each bin. The mass is then converted to volume dividing by the species density, and the average refractive index in each size bin is calculated using the species' volume as weighting factor. For the internal homogeneous assumption (HOM), the volume-weighted average is over all species, while for the core-shell assumption (CSBC and CSBCV), the refractive index is calculated for a core (black carbon only in this study) and for a homogeneously mixed shell (all non-black carbon species). Mie calculations out using the code based on Mishchenko et al. (1999) for external and homogeneous internal mixing, and the code based on Toon and Ackerman (1981) for the core-shell internal mixing. For some extreme situations, such as very small or zero core size, the code do not attempt to perform extrapolations and returns a failed calculation. Depending on the combination of aerosol species, the number of valid calculation may thus slightly vary (see Tables 5 and 6).

We further distinguish the core-shell calculation into two cases, that differ in the procedure used to combine the log-normal distributions in a single distribution (needed for the calculation of the volume average refractive index). In the CSBC case, the size distribution of each species is left unchanged, while in the CSBCV case, a single size distribution is calculated before the size-dependent refractive index calculation. The basic difference of the two cases is in the resulting core mass fraction as a function of particle size, as illustrate in Figure 4. In the CSBC case, the core fraction varies smoothly from 1 for small particles

to 0 for large particles, while the CSBCV case is equivalent to assuming a single volume-average core fraction for all sizes. As previously noted in Figure 2 and Figure 3, the core mass fraction is inversely proportional to the rBC ratio (eq. 5), which is in turn proportional to the single scattering albedo and the core absorption enhancement. Therefore, it is relevant how the combination of size distributions is carried out. While the sum of log-normals is straightforward (CSBC case), the calculation of a single size distribution is more complex, and is carried out as follows. First, the particles average volume is computed for each species $i$:

$$v_i = \frac{4}{3}\pi r_{g,i}{}^3 e^{4.5(\log \sigma_{g,i})^2} \tag{6}$$

Second, the total volume concentration of each species is computed:

$$V_i = \frac{M_i}{\rho_i} \tag{7}$$

Third, the volume-average standard deviation and particle volume are calculated as:

$$\langle \sigma_g \rangle = \frac{\sum_{i=1}^{n} \sigma_{g,i} V_i}{\sum_{i=1}^{n} V_i} \tag{8}$$

$$\langle v \rangle = \frac{\sum_{i=1}^{n} v_i V_i}{\sum_{i=1}^{n} V_i} \tag{8}$$

Finally, the single, volume-average, mean radius is calculated as:

$$\langle r_g \rangle = \left(\frac{3}{4\pi}\langle v \rangle e^{-4.5(\log\langle\sigma_g\rangle)^2}\right)^{1/3} \tag{9}$$

Since in the real atmosphere a combination of externally and internally mixed particles is typically found, we also test for the absorption properties in case of partial internal mixing (PIM) of particles. This is carried out using two simple empirical parameterizations of aerosol aging reported by Cheng et al. (2012), in order to calculate for each scene the fraction of internally mixed particles (*Fin*). The first parameterization is based on the fraction of oxidized nitrogen oxides (NOz = NOy – NOx) on total reactive nitrogen (NOy = PAN + HNO₃ + N₂O₅):

$$Fin = 0.572 + 0.209 \frac{[NOz]}{[NOy]} \tag{10}$$

The second is based on the rBC ratio:

$$Fin = 0.522 + 0.0088 \frac{[SIA]+[OC]}{[BC]} = 0.522 + 0.0088\, rBC \tag{11}$$

The two partial internal mixing cases (PIM-NOx and PIM-rBC in Table 4) combine the EXT external mixing case and the CSBC core-shell case. The aerosol optical properties are calculated as the external mixing of the two cases, weighted by *Fin*.

## 3 Results

The aim of the work is to estimate an observational constraint on the modelling of absorption of solar radiation by black carbon, in particular the absorption enhancement expected for internally mixed BC with respect to externally mixed BC. The comparison of AQMEII-3 simulations (see section 2.2) with aerosol optical quantities retrieved from AERONET sunphotometers network is thus limited to scenes classified as dominated by black carbon ("BC") or black and brown carbon

("BC+"BrC"), i.e. discarding those dominated by dust (see section 2.1 for details). We inter-compare absorption properties calculated in FlexAOD sensitivity tests with varying aerosol mixing state assumptions, as described in section 2.3 and summarized in Table 4. Results based on FlexAOD calculations using aerosol species profiles combined across all regional scale models are presented in tables and figures of the manuscript, while results for the same FlexAOD calculations performed

separately for the aerosol species profiles provided by each individual model are given in the supplementary online material. We generally found an underestimation of the aerosol optical depth at 440 nm $\tau_{440}$ of ~60% (Figure S7 and Table S2), and the bias is almost the same for all sensitivity tests, reflecting the fact that $\tau$ is primarily determined by the underlying aerosol mass and only secondarily affected by the mixing state. Indeed, the internal mixtures distribute the same aerosol mass in less numerous but larger particles with respect to external mixing, and the two effects compensate resulting in roughly the same

optical depth. The underestimation of $\tau$ reflects a general underestimation of PM2.5 concentrations, but it may also denote a potential bias in the static size distributions assigned to the species in FlexAOD. As illustrated by Obiso and Jorba (2018), $\tau$ is sensitive to the assigned size distributions, in particular to the standard deviation $\sigma_g$. In particular, the optical depth may be altered by a factor of 2 or more with a 20% change of the log-normal parameters. This implies that refining the FlexAOD parameters (Table 3) might reduce the bias of the calculated $\tau$, but this is beyond the scope of the current manuscript.

We also found a 20-30% underestimation of the scattering Ångström exponent $SAE_{675}^{440}$ (eq. 3) calculated by FlexAOD (Figure S11 and Table S4), denoting that scattering efficiency is decreasing with increasing wavelength at a lower rate than AERONET observations. A lower $SAE$ is associated with larger particles, implying that the assigned size distributions result in slightly larger particles than those retrieved by the AERONET inversion. Indeed, the underestimation of $SAE$ is larger for internal mixing compared to external mixing, because the size of the particles is larger in the former case. Again, this bias could

potentially be addressed by refining the FlexAOD log-normal parameters, but this is not the intent of this study.

Instead, our focus is on the simulated absorption properties, that vary little with changing log-normal size parameters (e.g. Obiso and Jorba, 2018). However, in order to avoid confusion in the interpretation of results, we restrict the subsequent analysis to scenes where the mass and the size of the particles are reasonably simulated by the models. To this end, we discard all scenes where the difference of volume concentration and effective radius between AERONET retrievals and FlexAOD

simulations is larger than a factor of 2. This reduces the size of the dataset to about 10% of the original.

In Figure 5 and Table 5 we present the comparison of the single scattering albedo at 440 nm $\omega_{0,440}$ between AERONET retrievals and AQMEII-3 simulations, for the different sensitivity tests on aerosol mixing state. We found that under external mixing assumption the models tend to overestimate $\omega_{0,440}$ by 0.03-0.04 (3-5%), while they tend to underestimate it under internal mixing assumptions. It should be noted that, although the relative bias is apparently low, it is comparable in magnitude to the dispersion of the data (the standard deviation is 0.06 and 0.12 over Europe and North America, respectively). The CSBC

case has a negative bias of the same order of magnitude as the EXT case, while the HOM and CSBCV cases have a relative bias a factor of 3 higher (-12/-15%). This is consistent with previous findings that the homogeneous internal mixing is unphysical, because perfect stirring of black carbon inside a particle is impossible, and exaggerates the BC absorption enhancement (Bond et al., 2006, 2013). The CSBCV case, which yields results similar to HOM, uses a single volume-average

size distribution instead of the sum of the individual distributions (as done in CSBC), and therefore it has a value of the core mass fraction that is constant with the particle size. This points out that accounting for variations of the core mass fraction with particle size is important in terms of resulting single scattering albedo (Fierce et al., 2017). Interestingly, the smallest $\omega_{0,440}$ bias is found for the partial internal mixing cases (PIM-*), which underestimate AERONET retrievals by 1-3% on average. This supports the initial hypothesis that a combination of external and core-shell internal mixtures should yield a more realistic representation of the real-world aerosol absorption in the atmosphere. Using a static factor $Fin$, Zhang et al. (2011) and Zhuang et al. (2013) also suggested that the partial internal mixing approach has the potential for a more realistic representation of aerosol radiative effects.

In Figure 6 we show the individual model $\omega_{0,440}$ normalized bias averaged over the selected AERONET scenes, for both Europe and North America. The overestimation of 3-5% in the EXT case is present in most models, with the exception of IT2, which is almost unbiased, and US3, which has a larger bias of ~10%. For IT2, the reason resides in the peculiar profile of BC noted in Figure 2, which is simulated at higher relative concentrations (denoted by low values of $rBC$) in the free troposphere with respect to other models. This results in values of $\omega_{0,440}$ of 0.7-0.8 in upper layers even when external mixing (i.e. no absorption enhancement) is assumed. On the other hand, the US3 model predicts relatively low concentrations of BC in the free troposphere, and this fact, combined with the lower values of $\omega_{0,440}$ in North American scenes (~0.82) with respect to European (~0.91), determines the larger bias. Regarding the internal mixing cases, all models have large negative biases in the HOM and CSBCV tests, while the bias is roughly halved in the CSBC test. Some models (DE1, ES1, and US3) have very small bias in the CSBC case, improving over the EXT case. These models has some of the largest BC absorption enhancement values $E_{abs}$ (between 2 and 3 throughout the vertical profile) of the ensemble. The two partial internal mixing cases (PIM-*) give generally very similar results, suggesting that the parameterization is quite robust despite being based on different proxies for the aging of particles (gas phase vs. aerosol phase). The resulting bias with respect to observations is the lowest of all cases in many models, specifically for DK1, ES1, FI1, and FRES1.

The calculated BC absorption enhancement in the internal mixing cases is always on average greater than the maximum value of ~1.5 suggested by Bond et al. (2006). This is illustrated in the profiles of Figure 2 and Figure 3, and summarized in Figure 7, which shows the column average of the $E_{abs}$ at 440 nm. In the CSBC case, most models have an average $E_{abs,440}$ in the range 1.8-2.5, with two models (DE1 and ES1) having $E_{abs,440} > 2.5$. For HOM and CSBCV cases, $E_{abs,440}$ is higher than 3 for most models, in particular over Europe, and more than 6 in one extreme case (DE1 in the HOM case). Although still higher than values recommended by Bond et al. (2006), the CSBC case is the one getting closer and partial internal mixing with EXT case would get the enhancement factor even closer. On the other hand, the HOM and CSBCV cases appear to predict too high and unrealistic $E_{abs}$ values.

Figure 8, Figure 9, and Table 6, examine the spectral variation of aerosol absorption properties through the absorption Ångström exponent $AAE_{675}^{440}$ (eq. 4). The theoretical value of $AAE_{675}^{440}$ is 1 for pure BC particles, such as those represented in an external mixture, while it varies in the range 0.6-1.3 for coated BC particles, such as those represented in core-shell internal mixing (Kirchstetter et al., 2004; Liu et al., 2017a). Values of $AAE_{675}^{440}$ of 1.5-2.0 or higher, denoting a more rapid decrease of

absorption with increasing wavelength, are related to the presence of BrC and dust (Russell et al., 2010; Wang et al., 2016). The observed $AAE_{675}^{440}$ average over selected AERONET scenes is $1.10 \pm 0.29$ and $1.19 \pm 0.28$ over Europe and North America, respectively, possibly indicating a predominant influence from BC and coated BC. All simulations tend to overestimate those values in all sensitivity tests, suggesting a more important influence of BrC in the simulations compared to observations. The model bias is generally larger over the North American domain (NMB 12-149%) than over Europe (NMB 9-74%). The lowest bias is found in the HOM and CSBCV cases, while all other cases have a similar and much higher positive bias.

The reason for the apparently good performance of HOM and CSBCV cases in simulating $AAE_{675}^{440}$ is explained by the different amplitude of the BC absorption enhancement $E_{abs}$ at 440 and 675 nm. In Figure 10 we show the ratio of $E_{abs,675}$ and $E_{abs,440}$ averaged over the selected AERONET scenes. The ratio is well above 1 for the HOM and CSBCV cases, while it is around 1, and for most models below 1, for the CSBC case. Considering that it is expected that $E_{abs}$ decreases with increasing wavelength (Liu et al., 2017a), a physically reasonable value of the ratio $E_{abs,675}/E_{abs,440}$ should be below 1. This observation suggests that HOM and CSBCV cases have good skills in reproducing the retrieved $AAE_{675}^{440}$, but for the wrong reason. Overall, among the internal mixing cases explored here, the CSBC case seem to be the one showing best promises for a physically sound simulation of the spectral absorption characteristics of atmospheric aerosol, although further testing and refinement on the underlying parameters is still needed.

Summarizing the comparison between the two continents, the selected AERONET observations generally show more absorbing (mean $\omega_{0,440}$ of 0.82 vs. 0.91) and spectrally dependent (mean $AAE_{675}^{440}$ of 1.19 vs. 1.10) aerosol over North America than Europe. The models broadly capture this variability, but display generally a larger bias over North America. The changes induced in the calculated optical quantities by the modifications tested here on the mixing state assumptions are consistent on the two regions.

### 3.1 Additional sensitivity tests on underlying assumptions

In this section, we expand the discussion on the underlying assumptions regarding physical and chemical properties of modelled aerosol optical properties, carrying out additional sensitivity tests using the FlexAOD tool. We apply the tests to one model, IT2, in order to reduce the computational time, selected as the one having the performance, in terms of $\omega_{0,440}$ and $AAE_{675}^{440}$, similar to the ensemble average of all models and not showing an outstanding bias for aerosol mass and composition. In particular, we shall focus our attention on the role played by BrC and the size distributions (see Table 3) in shaping the results illustrated above.

In Table 7 we list the description of the additional sensitivity tests which are discussed in this section. We run the tests in the two extreme and more physically relevant mixing assumption adopted above, i.e. external mixing (EXT) and core-shell (CSBC). The first subset of tests is related to the influence of the model bias in terms of aerosol species mass. From Table S1, we estimate that model IT2 overestimates sulfate by a factor of 3, ammonium and BC by a factor of 2, while nitrate and organic

fraction is in the range of observations. The tests 2-4 thus explore the effect of the mass adjustment on $\omega_{0,440}$ and $AAE_{675}^{440}$, as illustrated in the related scatterplot in Figure 11. The correction of secondary inorganic aerosol mass yields a negligible change in terms of calculated absorption properties, while the correction of BC mass introduces more change: the reduction of BC mass, as expected, reduces the absorption ($\omega_{0,440}$ increases) and makes its spectral variation more steep ($AAE_{675}^{440}$ increases).

The change is of the order of 3-4%, which is comparable to the magnitude of models' $\omega_{0,440}$ bias, but it is of the same sign and magnitude for external and core-shell mixing. The bias of BC mass is thus unlikely to alter the main conclusions regarding calculated absorption properties illustrate above.

The subsequent tests 5-6 are carried out to evaluate the effect of the assumptions made on aerosol size distributions. The first of this tests (GC), uses a completely different set of size distribution parameters. In particular, we substituted the log-normal

parameters of Table 3 with those used in the GEOS-Chem global chemistry transport model (http://wiki.seas.harvard.edu/geos-chem/index.php/Aerosol_optical_properties), as listed in Table 7. The result is a very little change in terms of absorption quantities, confirming that the results shown above are not very sensitive to the details of the assumed size distributions, in particular those regarding the material assumed to be in the shell.

In the second test devoted to size distributions (BC05), we modified only the size of BC. As shown in Table 3, the mean radius

of the BC size distribution is assumed to be 0.0118 μm, which is comparable to the size of a single spherule (monomer) of BC. As mentioned in section 2.3, in the real atmosphere the observed form of BC goes from fractal aggregates of monomers to more compact forms as it ages. We thus repeated the calculations with an increased mean radius of 0.5 μm, in the middle of the range of radiuses explored by Li et al. (2016). The effect in the external mixing case is a slight increase of the $\omega_{0,440}$ and increased variability of the $AAE_{675}^{440}$. In the core-shell case, both $\omega_{0,440}$ and $AAE_{675}^{440}$ decrease, implying that larger BC cores

increase the absorption and flatten its spectral dependence toward values more comparable with those deduced from AERONET measurements. As a caveat, the increase in the mean BC radius is what explains the difference between the CSBC and the CSBCV cases illustrated above. However, the $E_{abs}$ also increases by about 50% (not shown), thus a better simulation of $AAE_{675}^{440}$ is only apparently happening for the right reason, and this is certainly a point that should be further explored in future studies.

The final subset of tests 7-9 are devoted at exploring the role of assumptions made on the absorption properties of BrC. In the baseline sensitivity tests presented above, we adopted the extreme choice of assigning BrC characteristics to the primary organic fraction. However, also the primary fraction is generally a mix of white and brown aerosol (e.g. Laskins et al., 2015). In test BRC0, we switch off the absorption due to BrC, setting the imaginary part of primary OC to the low value of $10^{-8}$. The effect is a decreased absorption, denoted by the increase of $\omega_{0,440}$. More remarkably, there is a complete suppression of the

spectral dependence of the absorption, denoted by the flattening of the simulated $AAE_{675}^{440}$ values. In the case of external mixing, $AAE_{675}^{440} \sim 1$, with very little variability, which is consistent with the presence of only externally mixed BC as an absorber (Liu et al., 2017a, Liu and Mishchenko, 2018). In the case of core-shell, most of the variability is also suppressed, but the mean value of $AAE_{675}^{440}$ is around 1.4, denoting the absorption amplification $E_{abs}$ by the shell around BC. According to recent

calculations reported by Luo et al. (2018), the core-shell model is expected to exaggerate this amplification especially at shorter wavelengths, thus artificially increasing the calculated $AAE_{675}^{440}$.

In test 8 (BRCS), we swapped the role of primary and secondary organic carbon as radiation absorber. The results are generally similar to the reference case, but there is an increased variability in the simulated values, reflecting the secondary nature of the aerosol, which is photochemically produced down-wind of the sources, and thus generally more variable. In the last test 9 (BRCSH), we further suppressed the hygroscopic growth assumed for the secondary organic fraction, while the primary was assumed hydrophobic in all the tests carried out in this study. The absence of water uptake by the aerosol increases the absorption (indeed water has a refractive index of 1.32-1.35 in the visible and it does not absorb light significantly), but it does not affect much the its spectral variation.

Overall, the additional sensitivity tests allow to confirm the broad messages carried out in the first part of the result section, in particular regarding the differences in simulated optical properties with different mixing state assumptions. Moreover, they indicate main directions of refinement and improvement of the calculations, which we may summarize suggesting the introduction in future work of more details on: (1) the varying BC structure and size distribution, (2) the BrC aging and source-specific refractive index. Moreover, it looks to be appropriate the use of algorithms for the solution of the internal mixing problem more accurate than the core-shell model, for example the multiple-sphere T-matrix method.

**4 Conclusions**

Tests were carried out on the sensitivity of aerosol absorption in the visible spectrum to assumed mixing state using a suite of continental scale air quality simulations over Europe and North America and a stand-alone post-processing tool. The model results analysed are part of the third phase of the Air Quality Model Evaluation International Initiative (AQMEII, http://aqmeii.jrc.ec.europa.eu/, Galmarini et al., 2017). A single post-processing tool (FlexAOD, Curci et al., 2015, http://pumpkin.aquila.infn.it/flexaod/) has been used to derive aerosol optical properties from simulated aerosol speciation profiles. We compared calculations with one year of AERONET sunphotometer retrievals in order to identify the mixing state configuration that better reproduces the observed single scattering albedo and its spectral variation. The focus was on carbonaceous aerosol, in particular on the absorption enhancement of black carbon, expected when it is internally mixed with more scattering material. We carried out the comparison discarding AERONET scenes dominated by dust (the other important absorbing agent in atmospheric aerosol), and having a difference between simulated and observed aerosol volume concentration and effective radius larger than a factor of two.

When the particles are assumed to be externally mixed (EXT case), the single scattering albedo at 440 nm is overestimated by 0.03-0.05 (3-5%) on average, and the decrease of absorption efficiency with increasing wavelength (measured here with the absorption Ångström exponent between 440 and 675 nm) is overestimated by ~60% over Europe and ~150% over North America. The percent difference of the single scattering albedo with respect to the observations is on the same order of

magnitude as the standard deviation of the data (0.06 and 0.12 over Europe and North America, respectively). When the optical properties are calculated assuming a BC core coated with a shell made by all other species considered (primary organic, and secondary inorganic and organic aerosol; CSBC case), $\omega_{0,440}$ is underestimated by ~0.04 (4%) on average, and $AAE_{675}^{440}$ is overestimated by ~70% and ~100% over Europe and North America, respectively.

We tested two simple empirical parameterizations of aerosol aging (one based on the degree of oxidation of nitrogen oxides, the other based on the ratio of BC and other species mass) to combine the two calculations in a partial internal mixing configuration (PIM-NOx and PIM-rBC cases). Interestingly, the two parameterizations yield very similar results and the bias on $\omega_{0,440}$ is reduced to -1/-3%. The bias on $AAE_{675}^{440}$ is also in between the external and core-shell case, and thus still positively biased by 70-120%.

The spectral dependence of absorption derived from AERONET observations result in values of $AAE_{675}^{440}$ of $1.10 \pm 0.29$ and $1.19 \pm 0.28$ over Europe and North America, respectively, consistent with values expected in BC-dominated scenes. We found that two additional sensitivity tests reproduced these values with a positive bias of only 10-20%. One test assumes internal homogeneous mixing of all species (HOM case), but this configuration should be considered unrealistic, since BC cannot be well mixed with other material (Bond et al., 2013); moreover, it underestimates $\omega_{0,440}$ by ~14% (a bias three times larger than

other tests mentioned earlier). The other test is a core-shell configuration, but a single size distribution is assigned to all species, calculated from the volume average of the individual species' size distributions (CSBCV case). This test gives results very similar to the homogeneous mixing case, but is physically plausible. The methodology adopted to combine several size-distributions in a homogeneously mixed shell is thus a point that deserves further analysis in the future.

A qualitative investigation of BC absorption enhancement revealed that the CSBC case predicts $E_{abs}$ values at 440 nm mostly
in the range 1.8-2.5, while the HOM and CSBCV cases yield $E_{abs} > 3$. These values are higher than the limit of ~1.5 suggested by Bond et al. (2006), but the combination of EXT and CSBC in partial internal mixing has the potential of lowering the simulated $E_{abs}$ to values similar to this upper limit. Moreover, we found that $E_{abs}$ is increasing with wavelength (from 440 to 675 nm here) in the HOM and CSBCV cases, and this explains the apparently good performance of these tests in reproducing the observed $AAE_{675}^{440}$. However, experimental data suggest that $AAE_{675}^{440}$ should decrease with wavelength (a fact that is
confirmed by most models in the CSBC case), and thus HOM and CSBCV tests might be predicting a correct spectral dependence of the aerosol absorption for the wrong reason.

In conclusion, this work suggests that the combination of external and core-shell mixing state have the potential for a realistic representation of atmospheric aerosol absorption and its spectral dependence. However, the validation of model calculations using only sunphotometers retrievals as term of comparison is not exhaustive. Further evaluations against more comprehensive
campaign data that include a full characterization of the aerosol profile in terms of chemical speciation, mixing state, and related optical properties (such as in the study recently reported by Wang et al. (2017)) is certainly desirable. Moreover, the use of explicitly simulated aerosol size distributions should be included in future work, as opposed to the use of assigned size distributions as done here, in order to further investigate the effect of core mass fraction changing with aerosol size. The

introduction of more detailed treatment of the aging structure of BC and BrC is also recommended, in combination with algorithms more accurate than the core-shell model, such as the multiple-sphere T-matrix method.

**List of acronyms and symbols**

| | |
|---|---|
| AERONET | Aerosol robotic network |
| AQMEII | Air Quality Model Evaluation International Initiative |
| BC | Black carbon |
| BrC | Brown carbon |
| CCN | Cloud condensation nuclei |
| $E_{abs}$ | Black carbon absorption enhancement factor |
| ERFaci | Effective radiative forcing due to aerosol–cloud interactions |
| Fin | Fraction of internally mixed particles |
| FlexAOD | Flexible aerosol optical depth |
| OC | Organic carbon |
| REari | Radiative effect due to aerosol–radiation interactions |
| SIA | Secondary inorganic aerosol (sulfate, nitrate, ammonium) |
| AAE | Absorption Ångström exponent |
| $g$ | Asymmetry parameter |
| κ | Hygroscopicity parameter |
| λ | Wavelength |
| $m$ | Complex refractive index |
| $M$ | Aerosol mass concentration |
| $\omega_0$ | Single scattering albedo |
| ρ | Particle density |
| rBC | Black carbon ratio, rBC = ([SIA]+[OC]) / [BC] |
| $r_g$ | Geometric number mean radius |
| RH | Relative humidity |
| $\sigma_g$ | Geometric standard deviation |
| SAE | Scattering Ångström exponent |
| τ | Aerosol optical depth |
| $\tau_{abs}$ | Absorption aerosol optical depth |
| $\tau_{sca}$ | Scattering aerosol optical depth |
| $v$ | Single particle average volume |
| $V$ | Aerosol volume concentration |

## 5 Acknoledgements

Data analysis is carried out using R version 3.4.1 (R Core Team, 2017) and RStudio GUI version 0.98.1103, and packages rworldmap (South, 2011), ggplot2 (Wickham, 2009), and openair (Carslaw and Ropkins, 2012). The group from University of L'Aquila kindly thanks the EuroMediterranean Centre on Climate Change (CMCC) for the computational resources. P.T. is beneficiary of an AXA Research Fund postdoctoral grant. Contribution from CIEMAT was kindly supported by the Spanish Ministry of Agriculture and Fisheries, Food and Environment. The views expressed in this article are those of the authors and do not necessarily represent the views or policies of the U.S. Environmental Protection Agency. We thank two anonymous referees whose comments helped improving the robustness and clarity of the presented results.

**Appendix: definition of statistical indices used in the manuscript**

$$n = number\ of\ available\ data$$

$$O_i = i^{th}\ observation$$

$$M_i = i^{th}\ modelled\ value, paired\ with\ O_i$$

$$\bar{O} = mean\ observation = \frac{1}{n}\sum_{i=1}^{n} O_i$$

$$\bar{M} = mean\ modelled\ value = \frac{1}{n}\sum_{i=1}^{n} M_i$$

$$\sigma_O = standard\ deviation\ of\ observations = \sqrt{\frac{1}{n-1}\sum_{i=1}^{n}(O_i - \bar{O})^2}$$

$$\sigma_M = standard\ deviation\ of\ modelled\ values = \sqrt{\frac{1}{n-1}\sum_{i=1}^{n}(M_i - M)^2}$$

10 $$FAC2 = Fration\ of\ modelled\ values\ within\ a\ factor\ of\ two\ of\ observations = \frac{1}{n}\sum_{i=1}^{n}\left(i: 0.5 \le \frac{M_i}{O_i} \le 2.0\right)$$

$$MB = mean\ bias = \frac{1}{n}\sum_{i=1}^{n}(M_i - O_i)$$

$$NMB = normalized\ mean\ bias = \frac{MB}{\bar{O}}$$

$$RMSE = Root\ mean\ square\ error = \sqrt{\frac{1}{n}\sum_{i=1}^{n}(M_i - O_i)^2}$$

$$r = Pearson\ correlation\ coefficient = \frac{1}{n-1}\sum_{i=1}^{n}\left(\frac{O_i - \bar{O}}{\sigma_O}\right)\left(\frac{M_i - \bar{M}}{\sigma_M}\right)$$

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

**Tables**

**Table 1. List of AERONET sites selected for this study, over Europe and North America for the year 2010. Also reported are the counts of scenes classified as dominated by "Black Carbon" (BC), "Black Carbon + Brown Carbon" (BC+BrC), or "Dust", and the total number of available observations. The most frequent class for each site is highlighted in bold.**

| Site | Latitude | Longitude | BC | BC+BrC | Dust | Total |
|------|----------|-----------|-----|--------|------|-------|
| *Europe (20 sites)* | | | *4700 (60%)* | *1415 (18%)* | *1750 (22%)* | *7865 (393/site)* |
| Andenes | 69.28 | 16.01 | **123** | 10 | 30 | 163 |
| Barcelona | 41.39 | 2.12 | 205 | **309** | 112 | 626 |
| Brussels | 50.78 | 4.35 | **81** | 22 | 27 | 130 |
| Brujassot | 39.51 | -0.42 | **857** | 12 | 121 | 990 |
| Ersa | 43.00 | 9.36 | **151** | 12 | 29 | 192 |
| Huelva | 37.02 | -6.57 | 261 | 23 | **448** | 732 |
| Karlsruhe | 49.09 | 8.43 | **57** | 25 | 2 | 84 |
| Kyiv | 50.36 | 30.50 | **183** | 83 | 33 | 299 |
| Lecce University | 40.36 | 18.11 | **165** | 74 | 80 | 319 |
| Malaga | 36.72 | -4.48 | 317 | 89 | **448** | 854 |
| Messina | 38.20 | 15.57 | **189** | 17 | 93 | 299 |
| Moldova | 47.00 | 28.82 | **346** | 50 | 17 | 413 |
| Munich University | 48.15 | 11.57 | 31 | **153** | 16 | 200 |
| OHP Observatoire | 43.94 | 5.71 | **290** | 47 | 24 | 361 |
| Palencia | 41.99 | -4.52 | **90** | 26 | 23 | 139 |
| Salon de Provence | 43.61 | 5.12 | **69** | 17 | 1 | 87 |
| Sevastopol | 44.62 | 33.52 | **511** | 68 | 106 | 685 |
| Thessaloniki | 40.63 | 22.96 | **361** | 224 | 76 | 661 |
| Toravere | 58.26 | 26.46 | **132** | 29 | 25 | 186 |
| Toulon | 43.14 | 6.01 | **281** | 125 | 39 | 445 |
| *North America (9 sites)* | | | *1618 (64%)* | *398 (16%)* | *517 (20%)* | *2533 (281/site)* |
| Bozeman | 45.66 | -111.05 | **437** | 25 | 34 | 496 |
| BSRN BAO Boulder | 40.05 | -105.01 | **175** | 0 | 38 | 213 |
| Chapais | 49.82 | -74.98 | **68** | 28 | 2 | 98 |
| Easton Airport | 38.81 | -76.07 | **188** | 62 | 39 | 289 |
| Egbert | 44.23 | -79.75 | 21 | **202** | 121 | 344 |
| El Segundo | 33.91 | -118.38 | *133* | 28 | *127* | 288 |
| Halifax | 44.64 | -63.59 | **270** | 23 | 76 | 369 |
| Railroad Valley | 38.50 | -115.96 | **136** | 0 | 58 | 202 |
| Saturn Island | 48.78 | -123.13 | **190** | 22 | 22 | 234 |

**Table 2. Models from the phase 3 of the Air Quality Model Evaluation International Initiative (AQMEII) used by this study. Models' native grids and aerosol schemes differ, but output was remapped onto a common grid and the total mass of each aerosol component was used in this study. Please refer to section 2 for details.**

| ID | Domain | Model | Group | Grid Spacing | Aerosol model |
|---|---|---|---|---|---|
| DE1 | EU | COSMO-CLM CMAQ5.0.1 | Helmhotz-Zentrum Geesthacht (HZG) | 24 km x 24 km | Modal, 3 modes |
| DK1 | EU, NA | WRF DEHM | University of Aarhus | 50 km x 50 km | Modal, 2 modes |
| ES1 | EU | WRF/Chem | University of Murcia | 23 km x 23 km | Modal, 3 modes (MADE/SORGAM) |
| FI1 | EU | ECMWF-IFS SILAM | Finnish Meteorological Institute (FMI) | 18 km x 28 km | Sectional, 1-5 bins depending on the species |
| FRES1 | EU | ECMWF-IFS CHIMERE2013 | INERIS CIEMAT | 18 km x 28 km | Sectional, 8 bins |
| IT2 | EU | WRF/Chem3.6 | University of L'Aquila | 23 km x 23 km | Modal, 3 modes (MADE/VBS) |
| NL1 | EU | ECMWF-IFS LOTOS-EUROS | Netherlands Organization for Applied Scientific Research (TNO) | 36 km x 28 km | Modal, 2 modes |
| TR1 | EU | WRF3.5 CMAQ4.7.1 | Istanbul Technical University (ITU) | 30 km x 30 km | Modal, 3 modes |
| UK3 | EU | WRF3.4 CMAQ5.0.2 | University of Herfordshire | 18 km x 18 km | Modal, 3 modes |
| US3 | NA | WRF3.4 CMAQ5.0.2 | US Environmental Protection Agency (EPA) | 12 km x 12 km | Modal, 3 modes |

**Table 3. List of physical and chemical properties assigned to aerosol species. Ammonium has the same properties of sulfate. We assume spherical particles with log-normal size distribution, having geometric number mean radius $r_g$ and geometric standard deviation $\sigma_g$. Data source is Highwood (2009) for all, but growth factor, that uses Hess et al. (1998).**

| | Sulfate | Nitrate | Black carbon | Primary organic | Secondary organic |
|---|---|---|---|---|---|
| Particle density, $\rho$ (g cm$^{-3}$) | 1.769 | 1.725 | 1.8 | 1.47 | 1.3 |
| Refractive index at $\lambda$=550 nm, $m$ | $1.53 - i10^{-7}$ | $1.61 - i0.0$ | $1.85 - i0.71$ | $1.63 - i0.021$ | $1.43 - i0.0$ |
| Mean radius, $r_g$ ($\mu$m) | 0.05 | 0.065 | 0.0118 | 0.12 | 0.095 |
| Standard deviation, $\sigma_g$ | 2.0 | 2.0 | 2.0 | 1.3 | 1.5 |
| Growth factor at RH = 90% | 1.64 | 1.64 | 1.0 | 1.0 | 1.64 |

5  **Table 4. List of baseline sensitivity simulations on aerosol optical properties calculations. The case with full external mixing (EXT) is taken as reference, the other cases are sensitivity tests in which we changed one assumption per case related to the aerosol mixing state. The difference between CSBC and CSBCV cases is further illustrated in Figure 4. Results are shown in Figure 5-Figure 10.**

| Case | Label | Description | Mixing state | Mixing model | *Fin* method | Internal size distribution |
|---|---|---|---|---|---|---|
| 1 | EXT | Reference case, external mixing | External | - | - | - |
| 2 | HOM | Homogeneous internal mixing | Internal | Homogeneous volume average | - | Sum of log-normals |
| 3 | CSBC | Core-shell internal mixing, BC core | Internal | Core-shell | - | Sum of log-normals |
| 4 | CSBCV | Core-shell internal mixing, BC core, single volume-average size distribution | Internal | Core-shell | - | Volume-average log-normal (Figure 4) |
| 5 | PIM-NOx | Partial internal mixing of EXT and CSBC, weighted by NOx/NOy ratio | External and internal | External and core-shell | NOz/NOy ratio (eq. 10) | Sum of log-normals |
| 6 | PIM-rBC | Partial internal mixing of EXT and CSBC, weighted by rBC ratio | External and internal | External and core-shell | *rBC* ratio (eq. 11) | Sum of log-normals |

**Table 5. Comparison of FlexAOD modelled and observed single scattering albedo at 440 nm ($\omega_{0,440}$) in 2010 at AERONET stations over Europe and North America, only for scenes classified as "BC" or "BC+BrC"-dominated, and having a modelled aerosol volume concentration and effective radius within a factor of 2 of observations. Simulation labels are defined in Table 4 and statistical indices are defined in the Appendix. The number of data *n* may vary from case to case, due to numerical failures in the optical calculations.**

| Europe | n | $\bar{O}$ | $\bar{M}$ | $\sigma_O$ | $\sigma_M$ | FAC2 | MB | NMB | RMSE | r |
|---|---|---|---|---|---|---|---|---|---|---|
| 1.EXT | 3972 | 0.91 | 0.94 | 0.06 | 0.04 | 1.00 | 0.03 | 0.03 | 0.08 | 0.03 |
| 2.HOM | 3824 | 0.91 | 0.78 | 0.06 | 0.06 | 1.00 | -0.13 | -0.14 | 0.15 | 0.03 |
| 3.CSBC | 3956 | 0.91 | 0.87 | 0.06 | 0.05 | 1.00 | -0.04 | -0.04 | 0.09 | 0.00 |
| 4.CSBCV | 3719 | 0.91 | 0.80 | 0.06 | 0.06 | 1.00 | -0.11 | -0.12 | 0.14 | 0.02 |
| 5.PIM-NOx | 3863 | 0.91 | 0.89 | 0.06 | 0.04 | 1.00 | -0.02 | -0.02 | 0.08 | 0.00 |
| 6.PIM-rBC | 3068 | 0.91 | 0.88 | 0.06 | 0.04 | 1.00 | -0.03 | -0.03 | 0.08 | 0.02 |
| N. America | n | $\bar{O}$ | $\bar{M}$ | $\sigma_O$ | $\sigma_M$ | FAC2 | MB | NMB | RMSE | r |
| 1.EXT | 201 | 0.82 | 0.86 | 0.12 | 0.06 | 1.00 | 0.04 | 0.05 | 0.15 | -0.12 |
| 2.HOM | 202 | 0.82 | 0.71 | 0.12 | 0.06 | 1.00 | -0.11 | -0.13 | 0.17 | -0.02 |
| 3.CSBC | 202 | 0.82 | 0.78 | 0.12 | 0.05 | 1.00 | -0.04 | -0.05 | 0.14 | -0.06 |
| 4.CSBCV | 211 | 0.82 | 0.70 | 0.12 | 0.07 | 1.00 | -0.12 | -0.15 | 0.19 | -0.08 |
| 5.PIM-NOx | 200 | 0.82 | 0.80 | 0.12 | 0.05 | 1.00 | -0.02 | -0.02 | 0.14 | -0.08 |
| 6.PIM-rBC | 186 | 0.81 | 0.80 | 0.13 | 0.05 | 0.99 | -0.01 | -0.01 | 0.14 | -0.12 |

**Table 6. Same as Table 5, but for absorption Ångström exponent between 440 and 675 nm ($AAE_{675}^{440}$).**

| Europe | n | $\bar{O}$ | $\bar{M}$ | $\sigma_O$ | $\sigma_M$ | FAC2 | MB | NMB | RMSE | r |
|---|---|---|---|---|---|---|---|---|---|---|
| 1.EXT | 3972 | 1.10 | 1.78 | 0.29 | 0.45 | 0.78 | 0.68 | 0.61 | 0.86 | -0.01 |
| 2.HOM | 3824 | 1.10 | 1.20 | 0.29 | 0.08 | 0.95 | 0.10 | 0.09 | 0.31 | 0.00 |
| 3.CSBC | 3956 | 1.10 | 1.88 | 0.29 | 0.40 | 0.70 | 0.78 | 0.71 | 0.92 | 0.01 |
| 4.CSBCV | 3719 | 1.09 | 1.25 | 0.29 | 0.22 | 0.93 | 0.15 | 0.14 | 0.39 | 0.00 |
| 5.PIM-NOx | 3863 | 1.10 | 1.88 | 0.29 | 0.38 | 0.71 | 0.78 | 0.71 | 0.92 | 0.00 |
| 6.PIM-rBC | 3068 | 1.10 | 1.91 | 0.28 | 0.40 | 0.69 | 0.82 | 0.74 | 0.96 | -0.01 |
| N. America | n | $\bar{O}$ | $\bar{M}$ | $\sigma_O$ | $\sigma_M$ | FAC2 | MB | NMB | RMSE | r |
| 1.EXT | 201 | 1.19 | 2.96 | 0.28 | 0.93 | 0.37 | 1.77 | 1.49 | 2.00 | 0.08 |
| 2.HOM | 202 | 1.19 | 1.33 | 0.28 | 0.16 | 0.95 | 0.15 | 0.12 | 0.35 | 0.05 |
| 3.CSBC | 202 | 1.19 | 2.44 | 0.28 | 0.47 | 0.49 | 1.25 | 1.05 | 1.36 | 0.08 |
| 4.CSBCV | 211 | 1.18 | 1.44 | 0.28 | 0.31 | 0.94 | 0.26 | 0.22 | 0.48 | 0.05 |
| 5.PIM-NOx | 200 | 1.19 | 2.54 | 0.28 | 0.55 | 0.47 | 1.35 | 1.13 | 1.47 | 0.07 |
| 6.PIM-rBC | 186 | 1.20 | 2.61 | 0.28 | 0.54 | 0.42 | 1.41 | 1.18 | 1.54 | 0.04 |

**Table 7. List of additional sensitivity tests on BrC and size distribution assumptions. Here the changes are evaluated with respect to both the EXT and CSBC cases described in Table 4, changing one assumption per case. Results are shown in Figure 11.**

| Test | Label | Description | Change |
|------|-------|-------------|--------|
| 1 | REF | Same as the reference case EXT and CSBC | |
| 2 | AJSIA | Adjusted mass of secondary inorganic aerosol (SIA) | $SO_4 \times 0.33$<br>$NH_4 \times 0.5$ |
| 3 | AJBC | Adjusted mass of BC | $BC \times 0.5$ |
| 4 | AJ | Adjusted mass of both SIA and BC | $SO_4 \times 0.33$<br>$NH_4 \times 0.5$<br>$BC \times 0.5$ |
| 5 | GC | Size distribution from GEOS-Chem | Log-normal parameters:<br>$SO_4$) $r_g = 0.07$, $\sigma_g = 1.6$<br>$NO_3$) same as $SO_4$<br>BC) $r_g = 0.02$, $\sigma_g = 1.6$<br>Prim. OC) $r_g = 0.063$, $\sigma_g = 1.6$<br>Sec. OC) same as prim. OC |
| 6 | BC05 | Increased BC radius, in order to represent aged compact aggregate | $r_g = 0.5$ μm |
| 7 | BRC0 | Primary OC non absorbing | Imaginary part of primary OC refractive index = $10^{-8}$ |
| 8 | BRCS | Secondary OC absorbing | Refractive index of primary and secondary OC swapped |
| 9 | BRCSH | Same as BRCS, but without hygroscopic growth of secondary OC | Same as BRCS, plus hygroscopic growth of secondary OC = 1 |

## Figures

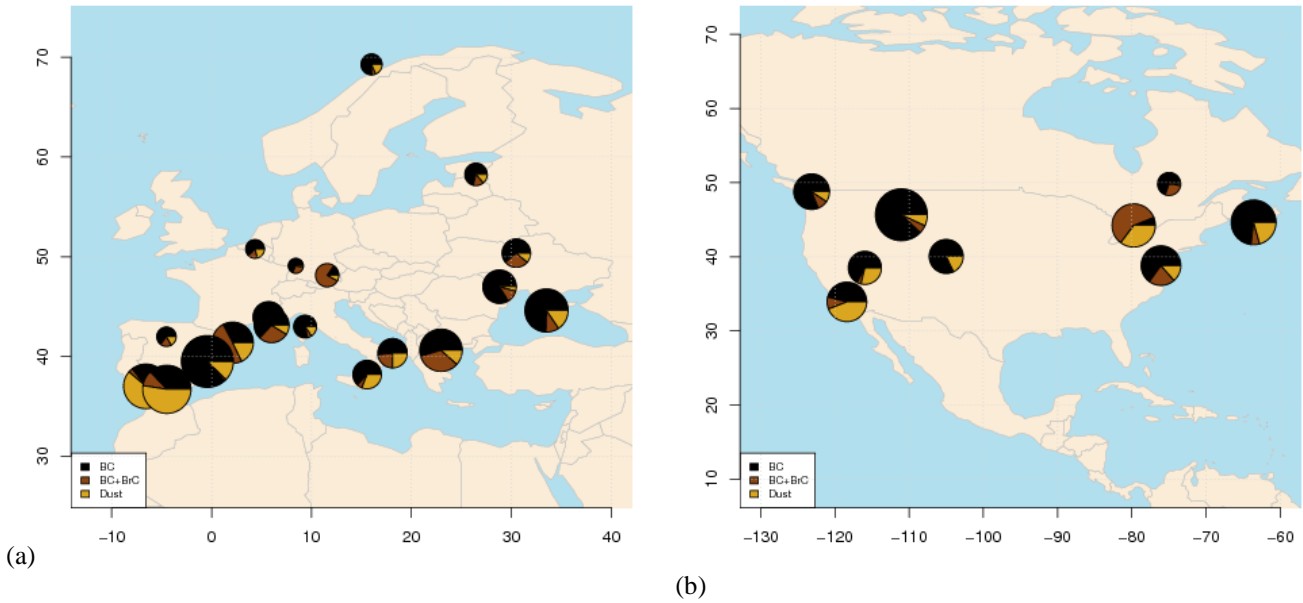

(a)

(b)

**Figure 1. Location of AERONET sunphotometer stations selected over (a) Europe and (b) North America. We use Level 2.0 inversion products for the year 2010, filled with Level 1.5 for scenes where absorption data (spectral single scattering albedo and absorption aerosol optical depth) were discarded in Level 2.0. The pie charts display the relative abundance of scenes classified as dominated by "Dust" (dark yellow), "Black Carbon" (black), or "Black Carbon + Brown Carbon" (brown). The size of the pies is proportional to the total number of observations.**

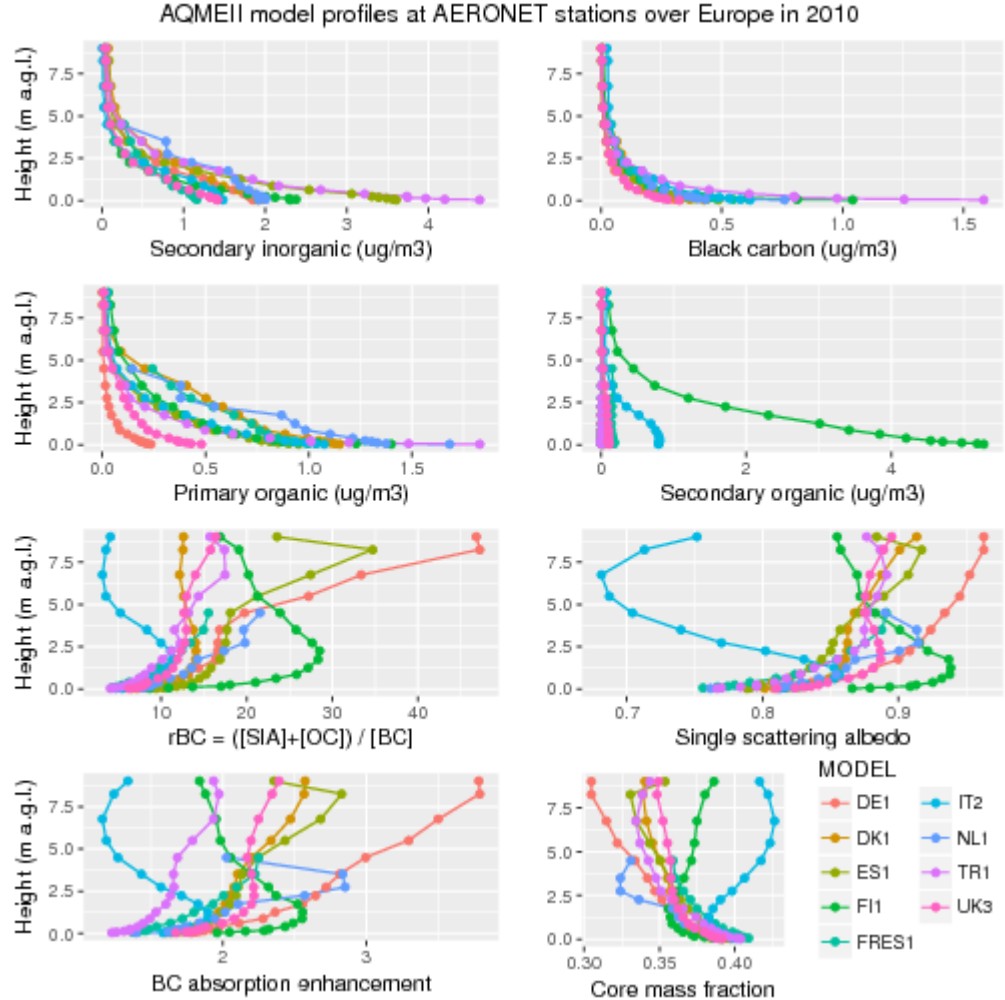

**Figure 2. Average model profiles sampled at locations and timings of AERONET observations available for the year 2010 over Europe. Top four panels show the simulated aerosol species concentrations included in the subsequent optical calculations. The ratio of total concentration of secondary inorganic aerosol (SIA) and organic carbon (OC, primary plus secondary) to black carbon (BC) also qualitatively illustrates the air mass chemical aging (larger for more aged aerosol). The single scattering albedo is that calculated using external mixing assumption (simulation EXT in Table 4). BC absorption enhancement is the ratio of absorption optical depths of simulation CSBC (core-shell internal mixing) to EXT. The core mass fraction is that calculated in the CSBC simulation.**

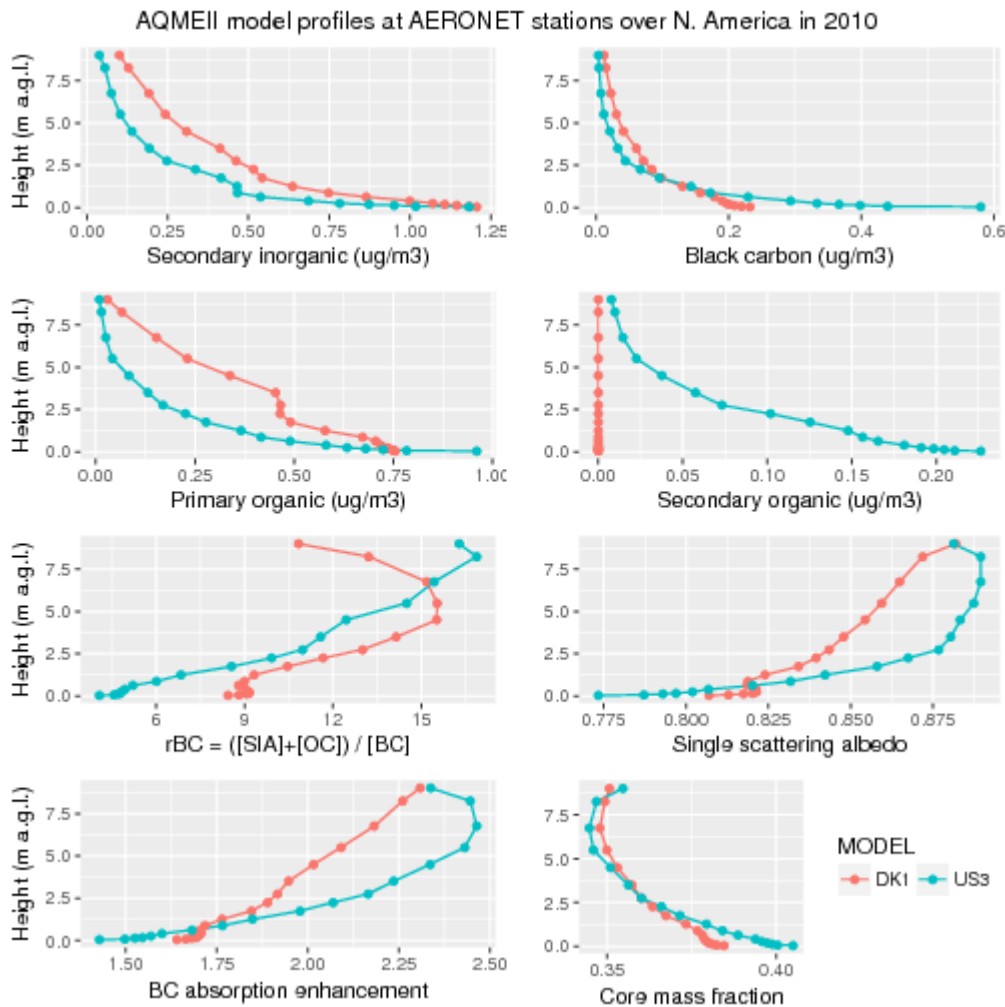

**Figure 3. Same as Figure 2, but for North America.**

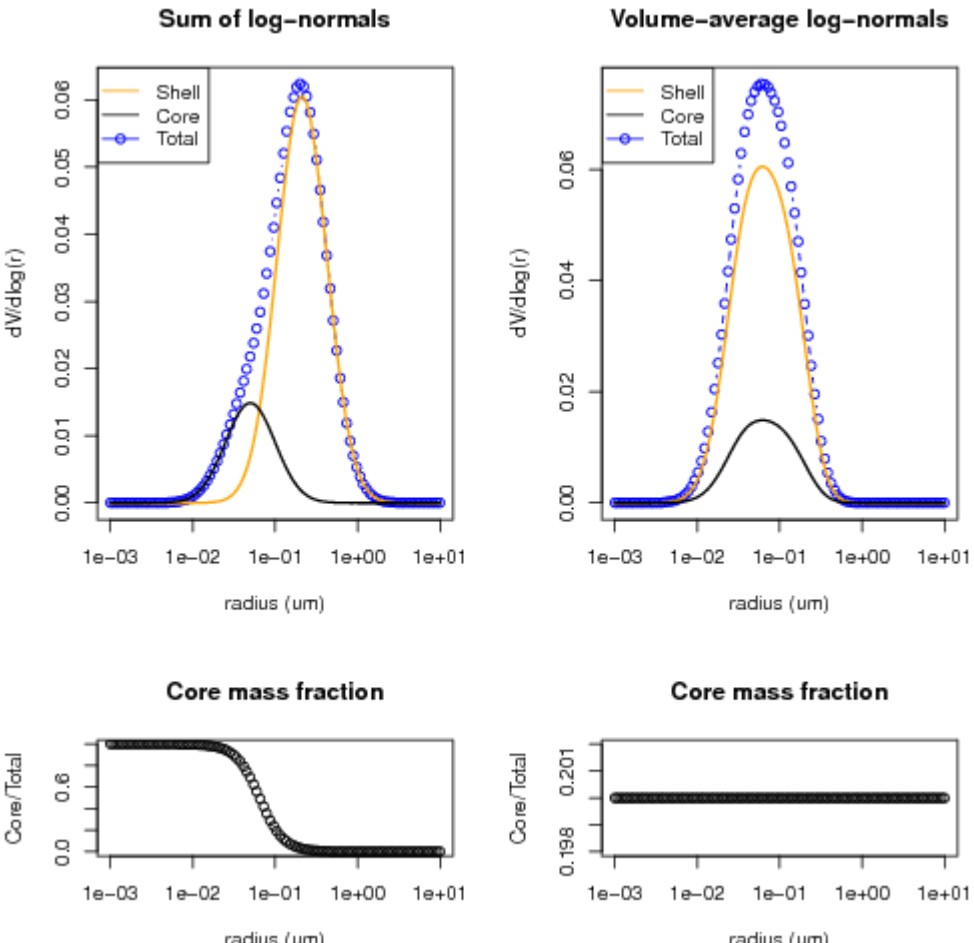

**Figure 4. Illustration of the different combination of size distributions tested in sensitivity simulations CSBC (left) and CSBCV (right). The size distributions of each species can be kept unchanged and summed in each size bin with the others (CSBC, left), or a single volume-average size distribution for all species can be computed (CSBCV, right). On the bottom panels, the resulting core mass fractions are shown as a function of particle radius. A lower core mass fraction is typically associated with higher core absorption enhancement.**

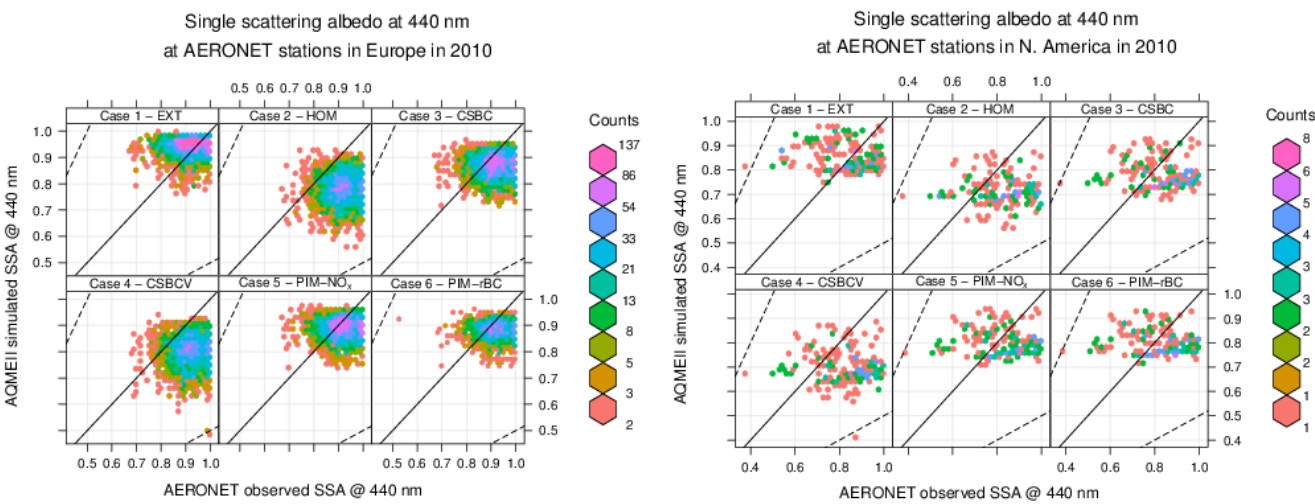

**Figure 5. Comparison of FlexAOD modelled and observed single scattering albedo at 440 nm ($\omega_{0,440}$) for the 2010 at AERONET stations over (a) Europe and (b) North America, only for scenes classified as "BC" or "BC+BrC"-dominated, and having a modelled aerosol volume concentration and effective radius within a factor of 2 of observations. Simulation labels are defined in Table 4.**

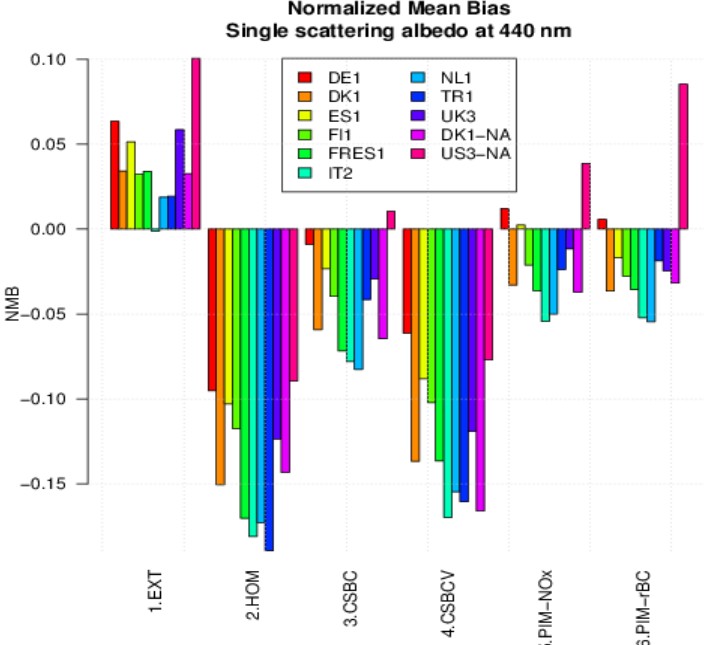

**Figure 6. Normalized mean bias of single scattering albedo at 440 nm ($\omega_{0,440}$) averaged over AERONET scenes for the year 2010.**

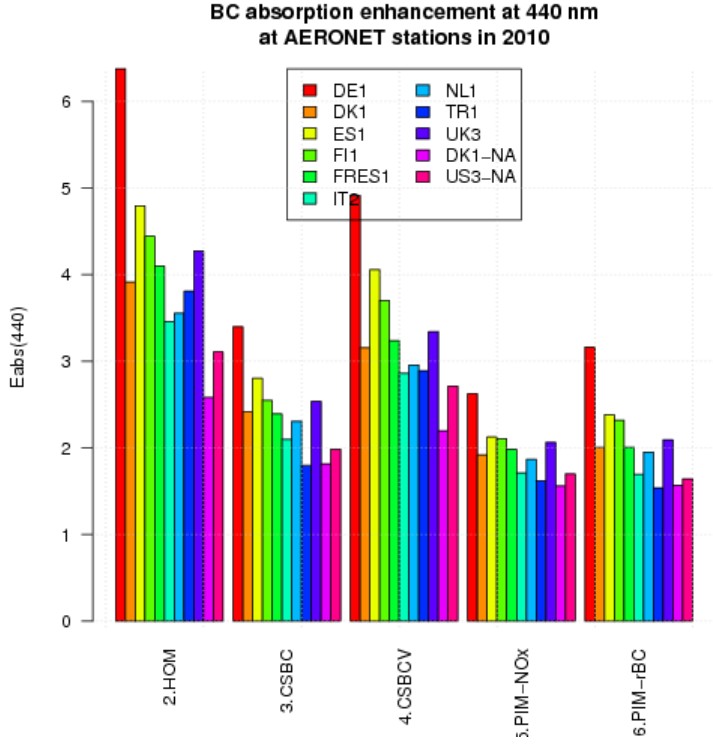

**Figure 7. Black carbon absorption enhancement at 440 nm ($E_{abs,440}$) averaged over AERONET scenes for the year 2010.**

(a)                                                    (b)

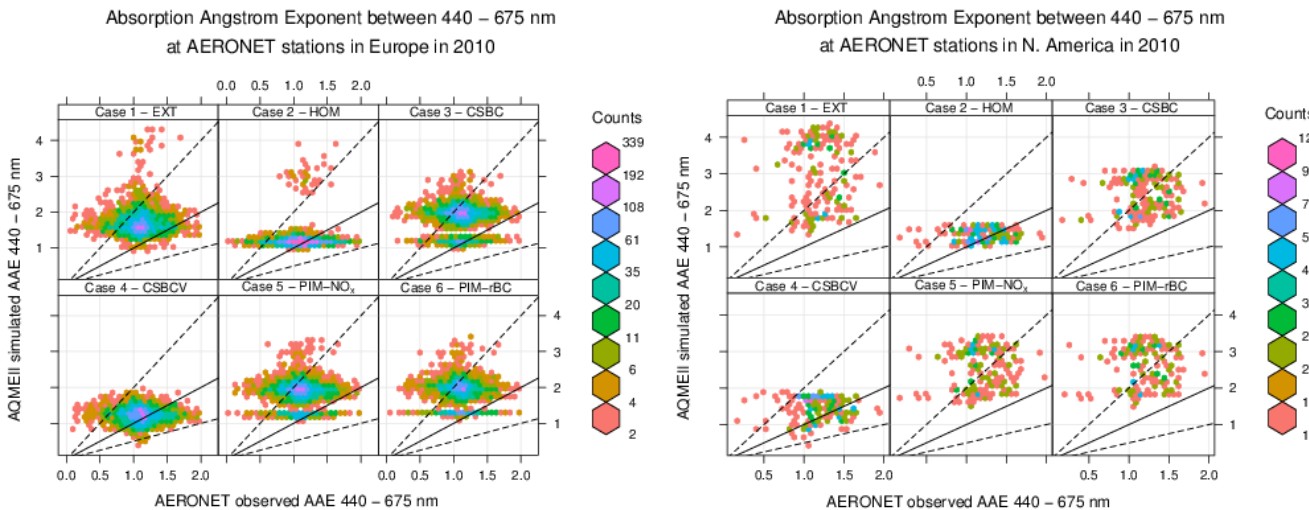

**Figure 8. Same as Figure 5, but for absorption Ångström exponent between 440 and 675 nm ($AAE_{675}^{440}$).**

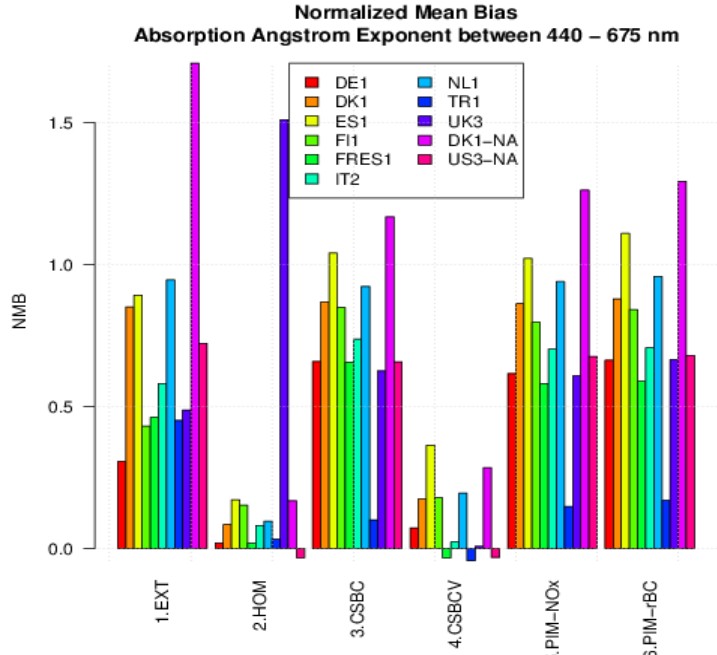

**Figure 9. Same as Figure 6, but for absorption Ångström exponent between 440 and 675 nm ($AAE_{675}^{440}$).**

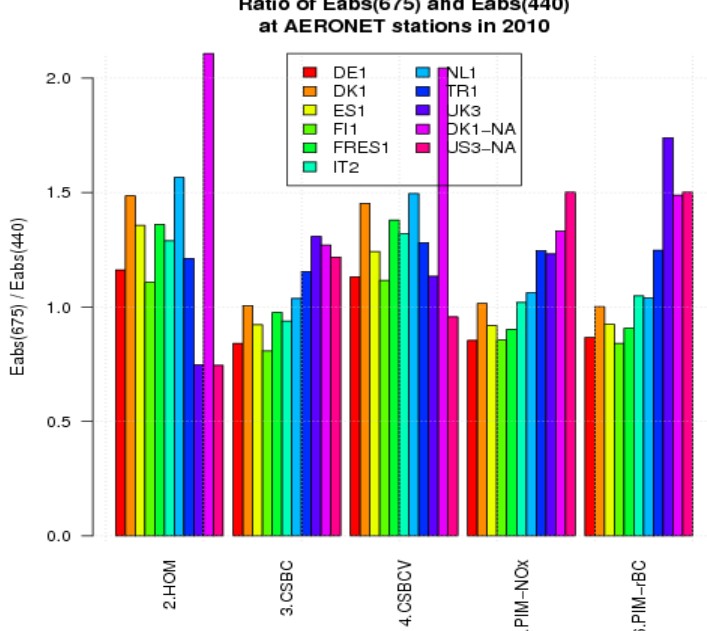

**Figure 10. Ratio of $E_{abs,675}$ and $E_{abs,440}$ averaged over AERONET scenes for the year 2010.**

(a)                                                                 (b)

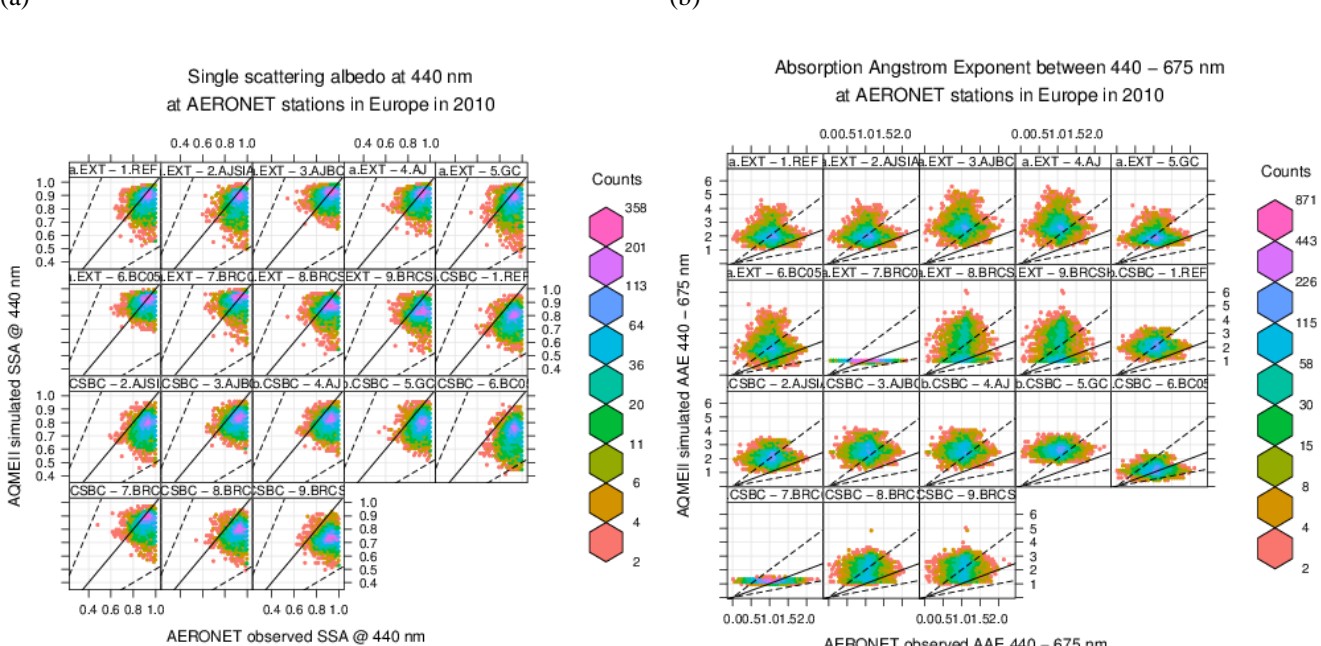

**Figure 11. Comparison of FlexAOD modelled and observed (a) single scattering albedo at 440 nm ($\omega_{0,440}$) and (b) absorption Ångström exponent between 440 and 675 nm ($AAE_{675}^{440}$) for the 2010 at AERONET stations, carried out with model IT2 for additional**
5   **sensitivity tests described in Table 7.**