# Peer review of "Modelling black carbon absorption of solar radiation: combining external and internal mixing assumptions"

_Atmospheric Chemistry and Physics, 2018_

## Referee Comment (RC1) · Anonymous Referee #2 · 7 May 2018

The paper presents a numerical study on the optical properties, especially absorption, of black carbon based on different assumptions on the mixing states and regional aerosol models. The AQMEII-3 results are used as the input aerosol properties, and a generalized off-line tool FlexAOD provides a unified method to give the optical properties, which are compared and evaluated by the AERONET results. By connecting the modeled aerosol properties and the observed optical properties, the manuscript presents a unique aspect to understand the absorption of BC aerosols. The paper is well designed and well organized, and I recommend it to publish in ACP after revision. The following lists my comments on the manuscript.

General comments:
1. As mentioned by the authors, the underestimation on total mass is the primary reason for the low AOD obtained, and this can neither be ignored on discussing the absorption properties of BC. This becomes critical because the BC concentration, which is also the primary factor for absorption estimation, may be significantly over- or underestimated. Considering the completely different profile results presented in Figures 2 and 3, the relative performances of the models would definitely influence the absorptions.
2. The modeled results are evaluated by comparing with the observations from AERONET measurements. However, the optical properties of the AERONET are retrieval products based on certain assumptions, and this means the results may differ if different assumptions were made for the retrieval. In other words, how would the uncertainties related to the AERONET observations themselves influence the evaluation of this study?
3. The mean radius of BC is assumed to be 11.8 nm based on Table 3, which is close to the size of monomer in BC aggregates. It is well known that the BC in the atmosphere is in the form of aggregates of those small monomers, and how would the non-spherical geometry influence the results. It would be difficult to account for the aggregation structures in such a work due to the computational burden, but it is worth to discuss the potential influences by considering previous studies. For example, Li et al. (https://doi.org/10.1002/2015JD024718) evaluated the influences of aggregation on BC optical properties especially AAE, and the effects of internal mixing was also studied by the same group (https://doi.org/10.1016/j.jqsrt.2016.10.023).
4. It is noticed that the primary organic aerosol is also absorptive. Is the influence also considered for estimating BC absorption? Its effects should also be removed for estimation on the absorption enhancement.

Specific comments:
1. "Grid Spacing" in Table 2 is listed as either km or degree, and it should be unified for better comparison.

2. The information in Table 4 is not well summarized, and the differences among the six models should be known with only reading the table.
3. In Figure 5, the colormaps for small and large count numbers are close, and clearer colormap is suggested.
4. It seems that Europe and N. America do not show too many differences on the conclusions related to the mixing state and absorption simulations. There are much less data for evaluating the N. America case, which makes the discussion less solid (e.g. Figs 5b and 8b). Why not just focus on Europe, because it will not change the conclusions of the manuscript but makes it much easier for discussion.
5. For Figures 6 and 9, are the values in the y-axis in unit of percentage? For example, does 1.0 mean 100% or 1%?

---

## Referee Comment (RC2) · Anonymous Referee #1 · 10 May 2018

General comments:

This work presents an interesting modelling study on the impact of mixing assumptions in the optical properties and radiative impact of aerosols in the atmosphere. The focus is on the absorption of Black Carbon (BC) and the impact of assuming external or internal mixtures. The authors use results from the regional models contributing to the third phase of the Air Quality Model Evaluation International Initiative (AQMEII), and through the use of an off-line tool to compute absorption properties (FlexAOD), model results are compared with AERONET sunphotometer retrievals.

The weaknesses of the study are: (1) the inconsistency of using prescribed microphys-

ical and optical properties for the aerosols different to the ones used by the models. The mass simulated by the set of models from AQMEII are strongly dependent on the microphysical properties of the aerosols assumed by each model (i.e., density, size distribution). The authors should justify that the harmonization applied in the microphysical properties of the aerosols has a second order impact on the understanding of the uncertainty associated with the mixing states assumed in coated BC particles. (2) A second point that should clarified in the manuscript is the treatment of the Brown Carbon (BrC). If mineral dust is excluded, BrC is the second most absorbing aerosol in the atmosphere, and the treatment in models should be specified. Some organic aerosols have absorbing properties depending on the emission source, others may experience a browning effect through its aging. Some specific discussion on how models deal with that is required to understand the distinction done between bright organic aerosol and BrC. This has a significant impact on the results and conclusions of the work. (3) The underestimation of the aerosol mass concentration at surface level and in the column observed in the AQMEII models introduces an important uncertainty on the results discussed in the manuscript. A lack of a comprehensive evaluation of the chemical composition of the aerosols is lying in the fundaments of the work. Thus, some discussion on that should be introduced in the revised manuscript and make it clear in the conclusions.

The paper is generally well-written. I recommend it to publish in ACP after the previous weakness/questions and following specific and technical comments are addressed.

Specific comments:

- Page 4 Line 9: What is the criterion followed to select the AERONET stations? Are only urban stations selected?

- Page 5 Line 4: Is not the 1.2 thershold for AAE too small for BrC? Lack and Cappa (2010) showed that AAE values, ranging from $1-1.6$, can be observed for internally mixed OC/BC particles and suggested that aerosols with AAE exceeding 1.6 should

be classified as BrC.

- Page 5 Line 6: What is the difference between your approach and Wang et al. (2016)? Would this affect the results of the work?

- Page 5 Line 22: What are the chemical boundary conditions used in the models?

- Page 6 Line 4: A justification of the different results over Europe of model ES1 and over US of model DK1 compared with the others would be useful. Differences are quite significant.

- Page 6 Line 8: There are observational networks in EU and USA that measure chemical composition of particles (i.e., EBAS, EMEP, IMPROVE). Why are not used here to quantify the errors on BC, organic mass, sulfate and dust concentrations? The work presented in the manuscript would benefit from a clear quantificaction of the errors of the models on the chemical composition of the aerosols, at least at surface level.

- Page 6 Line 12: Zhang et al. (2017) showed that some aircraft experimental data suggest that BrC constitute a significant part of absorbing carbonaceous aerosols, especially at high altitudes (> 5km). Some model disagreement with observations seems related to the treatment of BrC. The assumptions used in the present work related to BrC should be described in more detail.

- Page 6 Line 15: It would be useful to understand the reasons of the differences of ES1 and TR1 models predicting the secondary inorganic species. If emissions are the same ones for all the models, the differences should not be so large among models.

- Page 6 Line 21: Here again, there is a need to clarify how BrC is treated in the models. There is some evidence that some BrC comes from secondary organic aerosols (Laskins et al., 2015).

- Page 6 Line 31: It would be useful to include here the description of how BC absorption enhancement and BC core mass fraction are calculated. Right now, the definitions are in the caption of Figure 2.

- Page 7 Line 7: It seems that EU models reproduce the AOD for the wrong reasons considering the significant bias in surface PM2.5. This points to the need of an evaluation of the chemical composition of this PM2.5. What are the implications to the results obtained from the work? It can be expected than a different aerosol vertical distribution will considerably modify the findings of the work.

- Page 7 Line 27: Are the optical properties of the shell species computed as an homogeneous internal mixing?

- Page 8 Line 4: From the results, it turns out that the methodology to combine size distributions is a critical point. This should be remarked in the conclusion of the manuscript as an open issue that deserves further analysis.

- Page 9 Line 18: Would the use of the original models size distribution reduce this bias instead of using a uniform assigned distributions? The assumption of a bulk aerosol implies a loss of detail from the original model results.

- Page 9 Line 26: It would be interesting to present the statistics of the models for this final subset of data. It is expected that the bias on PM2.5 and AOD will be reduced and the confidence with the findings may be stronger.

- Page 11 Line 5: The treatment of BrC in the simulations is not clear. From Table 4, the use of an imaginary refractive index of 0.021 following Highwood et al. (2009) for POM is already quite absorbing. Tang et al. (2016) and Kirchstetter et al. (2004) suggest an imaginary value of 0.03 for BrC from biomass burning. It seems that the assumptions selected for the optical properties of POM contributes to a more absorbing aerosol than the other way around.

In Table 4, it seems there is a typo in the imaginary refractive index of POM, which is set to 0.21. Highwood et al. (2009) suggest a value of 0.021 at 550 nm for the organic aerosol.

- Page 11 Line 14: Results of combining external and internal mixtures (PIM cases)

perform quite similar to CSBC. It would be worth mention it here.

- Page 11 Line 15: A general discussion comparing the EU and US results would be interesting as closing paragraph.

- Page 11 Line 26: This last sentence should be included in the abstract to clarify that the models are compared under BC dominant conditions.

- Page 11 Line 30: What are the recommendations for the modellers to improve the absorption of they systems? There is still significant variability among models, but some insights from the results indicate internal coating or mixtures of external and internal coating are recommended approaches. This can be highlighted in the conclusions. A comment on the impact seen between CSBC and CSBCV is an important result of the work. Only the method used to derive a resulting distribution of an internal mixture is still a significant source of uncertainty.

- Page 23 Table 3: Are the AOD values presented here computed with FlexAOD? Why are there such differences compared with Table S1, where US3 reports an AOD at 440 nm of 0.05?

- Page 23 Table 4: There is a typo in the imaginary part of the refractive index of POM. The refractive index of POM is more representative of a pure BrC rather than a mixture of white organic aerosol and BrC. Nakayama et al. (2012) or Liu et al. (2013) suggest a refractive index of 1.486 - i 2.5e-5 at 550 nm for secondary organic aerosols. Concerning the growth factor, is only the growth factor at 90% considered? The growth factor starts to be non-negligible at 75% relative humidity. Some description on role of the growth factor on the resulting refractive index of an internal mixture would be relevant for the work. Why are BC and POM considered hydrophobic?

- Page 24 Table 6: Some clarification on the numerical failures of the optical calculations is needed.

Technical corrections:

- Page 2 Line 15: I suggest to move this last sentence to the last paragraph of the introduction.

- Page 3 Line 22: Replace "external or internal" by "external and internal".

- Page 5 Line 23: Delete the second point after the reference (Flemming et al., 2015).

- Page 7 Line 16: Check syntax of the references "(source of data Highwood (2009) and Hess et al. (1998))".

- Page 7 Line 29: Delete the point after the word "mixing".

- Page 8 Line 11: Finalize the sentence with "and particle volume are:".

- Page 9 Line 15: Include a closing point at the end of the paragraph.

- Page 22 Table 2: Complete the table with the "Aerosol model description" for the University of Aarhus WRF-DEHM model.

- Figure 2, 3, 5 and 8: The quality or resolution of the figures is low for final publication.

References:

Kirchstetter, T. W., Novakov, T., and Hobbs, P. V. Evidence that the spectral dependence of light absorption by aerosols is affected by organic carbon. Journal of Geophysical Research, 109 (2004).

Lack, D. A.; Cappa, C. D. Atmos. Chem. Phys. 2010, 10, 4207.

Laskin, A., Laskin, J. & Nizkorodov, S. A. Chemistry of Atmospheric Brown Carbon. Chem. Rev. 115, 4335–4382 (2015).

Liu, P., Zhang, Y., and Martin, S. T. Complex Refractive Indices of Thin Films of Secondary Organic Materials by Spectroscopic Ellipsometry from 220 to 1200 nm. Environmental Science & Technology, 47, 13594–13601 (2013).

Nakayama, T., Sato, K., Matsumi, Y., Imamura, T., Yamazaki, A., and Uchiyama, A.

Wavelength Dependence of Refractive Index of Secondary Organic Aerosols Generated during the Ozonolysis and Photooxidation of alpha-Pinene. Scientific Online Letters on the Atmosphere, 8, 119–123, (2012).

Tang, M., Alexander, J. M., Kwon, D., Estillore, A. D., Laskina, O., Young, M. A., Kleiber, P. D., and Grassian, V. H. Optical and Physicochemical Properties of Brown Carbon Aerosol: Light Scattering, FTIR Extinction Spectroscopy, and Hygroscopic Growth. The Journal of Physical Chemistry A, 120, 4155–4166, (2016).

Zhang, Y. et al. Top-of-atmosphere radiative forcing affected by brown carbon in the upper troposphere. Nat. Geosci. 10, 486–489 (2017).

---

## Short Comment (SC1) · 31 May 2018

This manuscript is generally well written, but there seems to be one important piece missing, which is related to BC particle structures. This study assumed spheres for externally mixed BC and core-shell coating structure for internally mixed BC. However, recent observations (e.g., China et al., 2015; Wang et al., 2017) have shown irregular fractal aggregates for externally mixed BC and various non-core-shell coating structures for internally mixed BC. Further modeling studies (e.g., Scarnato et al., 2013; He et al., 2015, 2016) have indicated a large variation in BC optical properties due to the observed complex fractal and coating structures. Thus, the assumptions in the current

manuscript may lead to uncertainty in the calculations of BC optical properties. Thus, I suggest that the authors include these recent studies and add some discussions on this important issue.

References

China, S., et al.: Morphology and mixing state of aged soot particles at a remote marine free troposphere site: Implications for optical properties, Geophys. Res. Lett., 42, 1243–1250, doi:10.1002/2014gl062404, 2015.

He, C., et al.: Variation of the radiative properties during black carbon aging: theoretical and experimental intercomparison, Atmos. Chem. Phys., 15, 11967-11980, doi:10.5194/acp-15-11967-2015, 2015.

He, C., et al.: Intercomparison of the GOS approach, superposition T-matrix method, and laboratory measurements for black carbon optical properties during aging, J. Quant. Spectrosc. Radiat. Transf., 184, 287–296, doi:10.1016/j.jqsrt.2016.08.004, 2016.

Scarnato, B. V., et al.: Effects of internal mixing and aggregate morphology on optical properties of black carbon using a discrete dipole approximation model, Atmos. Chem. Phys., 13, 5089–5101, doi:10.5194/acp-13-5089-2013, 2013.

Wang, Y., et al.: Fractal dimensions and mixing structures of soot particles during atmospheric processing, Environ. Sci. Technol. Lett., 4, 487-493, doi:10.1021/acs.estlett.7b00418, 2017.

---

## Author Comment (AC1) · 30 Oct 2018

Reviewer #2

Color legend:

- Reviewer's comments in black
- *Authors' responses in red italic*

The paper presents a numerical study on the optical properties, especially absorption, of black carbon based on different assumptions on the mixing states and regional aerosol models. The AQMEII-3 results are used as the input aerosol properties, and a generalized off-line tool FlexAOD provides a unified method to give the optical properties, which are compared and evaluated by the AERONET results. By connecting the modeled aerosol properties and the observed optical properties, the manuscript presents a unique aspect to understand the absorption of BC aerosols. The paper is well designed and well organized, and I recommend it to publish in ACP after revision. The following lists my comments on the manuscript.

*We would like to thank the reviewer for the insightful comments delivered on our work. We believe that addressing them made the results more robust and general with respect to the initial submission. We performed a number of additional sensitivity tests (see new section 3.1 in the manuscript) and comparison to measurements in order to clarify the detailed points, as illustrated below.*

General comments:

1. As mentioned by the authors, the underestimation on total mass is the primary reason for the low AOD obtained, and this can neither be ignored on discussing the absorption properties of BC. This becomes critical because the BC concentration, which is also the primary factor for absorption estimation, may be significantly over or underestimated. Considering the completely different profile results presented in Figures 2 and 3, the relative performances of the models would definitely influence the absorptions.

*In summarizing the results, we tried to limit the influence of model bias in terms of mass and size distribution filtering out scenes with a large bias in terms of effective radius and volume concentration in the final analysis. However, the suggestion of the reviewer may help to present the results in a more clear way, and we thus compiled the following table with correspondences of AERONET stations and monitoring stations providing aerosol speciation measurements in Europe and North America:*

| AERONET Site | Latitude | Longitude | Database | Site | Latitude | Longitude |
|---|---|---|---|---|---|---|
| *Europe (20 sites)* | | | | | | |
| Andenes | 69.28 | 16.01 | - | | | |
| Barcelona | 41.39 | 2.12 | EMEP | Montseny | 41.77 | 2.35 |
| Brussels | 50.78 | 4.35 | - | | | |
| Brujassot | 39.51 | -0.42 | - | | | |
| Ersa | 43.00 | 9.36 | - | | | |
| Huelva | 37.02 | -6.57 | - | | | |
| Karlsruhe | 49.09 | 8.43 | - | | | |
| Kyiv | 50.36 | 30.50 | - | | | |
| Lecce University | 40.36 | 18.11 | - | | | |
| Malaga | 36.72 | -4.48 | - | | | |
| Messina | 38.20 | 15.57 | - | | | |
| Moldova | 47.00 | 28.82 | EMEP | Leova II | 46.49 | 28.28 |
| Munich University | 48.15 | 11.57 | - | | | |
| OHP Observatoire | 43.94 | 5.71 | - | | | |
| Palencia | 41.99 | -4.52 | EMEP | Campisabalos | 41.28 | -3.14 |
| Salon de Provence | 43.61 | 5.12 | - | | | |
| Sevastopol | 44.62 | 33.52 | - | | | |

| | | | | | | |
|---|---|---|---|---|---|---|
| Thessaloniki | 40.63 | 22.96 | - | | | |
| Toravere | 58.26 | 26.46 | - | | | |
| Toulon | 43.14 | 6.01 | - | | | |
| *North America (9 sites)* | | | | | | |
| Bozeman | 45.66 | -111.05 | - | | | |
| BSRN BAO Boulder | 40.05 | -105.01 | IMPROVE | Rocky Mountain NP | 40.28 | -105.55 |
| Chapais | 49.82 | -74.98 | - | | | |
| Easton Airport | 38.81 | -76.07 | IMPROVE | Washington DC | 38.88 | -77.03 |
| Egbert | 44.23 | -79.75 | IMPROVE | Egbert | 44.23 | -79.78 |
| El Segundo | 33.91 | -118.38 | IMPROVE | San Gabriel | 34.30 | -118.02 |
| Halifax | 44.64 | -63.59 | - | | | |
| Railroad Valley | 38.50 | -115.96 | IMPROVE | Great Basin NP | 39.01 | -114.22 |
| Saturn Island | 48.78 | -123.13 | IMPROVE | Olympic | 48.01 | -122.97 |

*For Europe, a reasonable correspondence is found only for 3 stations, while in North America for most stations (6 on 9). We collected the available observations for the year 2010 at these stations and carried out the comparison with concentrations of aerosol species at the first model level. The results are summarized in the supplementary revised Table S1 and the new Figure S7, and briefly discussed in section 2.2 in a new paragraph:*

*"Additional indications about models skills are gathered from the comparison with PM composition measurements available near the AERONET stations, for which we have stored the simulated PM speciation profiles of AQMEII models. The comparison is carried out at 3 stations over Europe and 5 stations over North America, and results summarized in Table S1 and Figure S7. Over North America, the two models have yearly average values mostly within ±1 µg m-3. Over Europe, most values are also within the same range, but there is a tendency toward overestimation of inorganic secondary species (sulfate, nitrate, ammonium) and black carbon, and underestimation of the organic carbonaceous fraction."*

*Given the very limited number of stations, we haven't emphasized the comparison to more than an additional indication on models skills.*

*We then used this, although partial, information to evaluate the impact of aerosol mass model bias on the main conclusions, as reported in a paragraph of the new section 3.1:*

*"We run the tests in the two extreme and more physically relevant mixing assumption adopted above, i.e. external mixing (EXT) and core-shell (CSBC). The first subset of tests is related to the influence of the model bias in terms of aerosol species mass. From Table S1, we estimate that model IT2 overestimates sulfate by a factor of 3, ammonium and BC by a factor of 2, while nitrate and organic fraction is in the range of observations. The tests 2-4 thus explore the effect of the mass adjustment on $\omega_{0,440}$ and $AAE_{675}^{440}$, as illustrated in the related scatterplot in **Errore. L'origine riferimento non è stata trovata.**. The correction of secondary inorganic aerosol mass yields a negligible change in terms of calculate absorption properties, while the correction of BC mass introduces more change: the reduction of BC mass, as might be expected, reduces the absorption ($\omega_{0,440}$ increases) and makes its spectral variation more steep ($AAE_{675}^{440}$ increases). The change is of the order of 3-4%, which is comparable to the magnitude of models' $\omega_{0,440}$ bias, but it is of the same sign and magnitude for external and core-shell mixing. The bias of BC mass is thus unlikely to alter the main conclusions regarding calculated absorption properties illustrate above."*

2. The modeled results are evaluated by comparing with the observations from AERONET measurements. However, the optical properties of the AERONET are retrieval products based on certain assumptions, and this means the results may differ if different assumptions were made for the retrieval. In other words, how would the uncertainties related to the AERONET observations themselves influence the evaluation of this study?

*This is an interesting point, but difficult to address here. According to the documentation of the AERONET retrieval algorithm ([https://aeronet.gsfc.nasa.gov/new_web/Documents/Inversion_products_V2.pdf](https://aeronet.gsfc.nasa.gov/new_web/Documents/Inversion_products_V2.pdf)) the iterative inversion for the estimation of absorption aerosol properties relies on the minimization of the difference between observed and simulated radiances. The simulated radiances are calculated from a mix of spherical and non-spherical particles, divided in two log-normally distributed modes, fine and coarse. Moreover, the vertical distribution of the aerosol layer is assumed to be homogeneous. A clean way to make the comparison of AERONET inversion products with simulations such as the ones presented here, would be to change the retrieval process itself, trying to minimize the error of the simulated radiances using underlying aerosol types more similar to those having the properties listed in Table 3 of the present manuscript. However, this is a very demanding task which is clearly beyond the scope of this study.*

3. The mean radius of BC is assumed to be 11.8 nm based on Table 3, which is close to the size of monomer in BC aggregates. It is well known that the BC in the atmosphere is in the form of aggregates of those small monomers, and how would the non-spherical geometry influence the results. It would be difficult to account for the aggregation structures in such a work due to the computational burden, but it is worth to discuss the potential influences by considering previous studies. For example, Li et al. (https://doi.org/10.1002/2015JD024718) evaluated the influences of aggregation on BC optical properties especially AAE, and the effects of internal mixing was also studied by the same group ([https://doi.org/10.1016/j.jqsrt.2016.10.023](https://doi.org/10.1016/j.jqsrt.2016.10.023)).

*We believe this is an important point that we did not discussed properly in the first version of the manuscript. A test and a summarizing paragraph in the new section 3.1 was devoted to that:*

*"In the second test devoted to size distributions (BC05), we modified only the size of BC. As shown in **Errore. L'origine riferimento non è stata trovata.**, the mean radius of the BC size distribution is assumed to be 0.0118 μm, which is comparable to the size of a single spherule (monomer) of BC. As mentioned in section 2.3, the real atmosphere observed form of BC goes from fractal aggregates of monomers to more compact forms as it ages. We thus repeated the calculations with an increased mean radius of 0.5 μm, in the middle of the range of radiuses explored by Li et al. (2016). The effect in the external mixing case is a slight increase of the $\omega_{0,440}$ and increased variability of the $AAE_{675}^{440}$. In the core-shell case, both $\omega_{0,440}$ and $AAE_{675}^{440}$ decrease, implying that larger BC cores increase the absorption and flatten its spectral dependence toward values more comparable with those deduced from AERONET measurements. As a caveat, the increase in the mean BC radius is what explains the difference between the CSBC and the CSBCV cases illustrated above. However, the $E_{abs}$ also increases by about 50% (not shown), thus a better simulation of $AAE_{675}^{440}$ is only apparently happening for the right reason, but this is certainly a point that should be further explored in future studies."*

4. It is noticed that the primary organic aerosol is also absorptive. Is the influence also considered for estimating BC absorption? Its effects should also be removed for estimation on the absorption enhancement.

*We acknowledge that the description and impact of the treatment of BrC was not properly addressed in the original manuscript. Given the many uncertainties in the absorption properties of organic matter (OM), we preferred to focus the discussion on BC absorption. Regarding OM we made one extreme choice, i.e. we treat primary OM as BrC and secondary OM as non-absorbing. From model output delivered for the intercomparison, we only have POM and SOM (when simulated) total mass, without any tracking of the sources or the aging. We acknowledge, however, that the extent to which this assumption influences the*

*main conclusions is not clear and we assessed it with further sensitivity tests on assumption on the treatment of OM absorption properties.*

*The outcome of additional tests is summarized in these paragraphs of the new section 3.1:*

*"The final subset of tests 7-9 are devoted at exploring the role of assumptions made on the absorption properties of BrC. In the baseline sensitivity tests presented above, we adopted the extreme choice of assigning BrC characteristics to the primary organic fraction. However, also the primary fraction is generally a mix of white and brown aerosol (e.g. Laskins et al., 2015). In test BRC0, we switch off the absorption due to BrC, setting the imaginary part of primary OC to the low value of $10^{-8}$. The effect is a decreased absorption, denoted by the increase of $\omega_{0,440}$. More remarkably, there is a complete suppression of the spectral dependence of the absorption, denoted by the flattening of the simulated $AAE_{675}^{440}$ values. In the case of external mixing, $AAE_{675}^{440} \sim 1$, with very little variability, which is consistent with the presence of only externally mixed BC as an absorber (Liu et al., 2017b, Liu and Mishchenko, 2018). In the case of core-shell, most of the variability is also suppressed, but the mean value of $AAE_{675}^{440}$ is around 1.4, denoting the absorption amplification $E_{abs}$ by the shell around BC. According to recent calculations reported by Luo et al. (2018), the core-shell model is expected to exaggerate this amplification especially at shorter wavelengths, thus artificially increasing the calculated $AAE_{675}^{440}$.*

*In tests 8 (BRCS), we swapped the role of primary and secondary organic carbon as radiation absorber. The results are generally similar to the reference case, but there is an increased variability in the simulated values, reflecting the secondary nature of the aerosol, which is photochemically produced down-wind of the sources, and thus generally more variable. In the last test 9 (BRCSH), we further suppressed the hygroscopic growth assumed for the secondary organic fraction, while the primary was assumed hydrophobic in all the tests. The absence of water uptake by the aerosol increases the absorption (indeed water has a refractive index of 1.32-1.35 in the visible and it does not absorb light significantly), but does not affect much the its spectral variation."*

Specific comments:

1. "Grid Spacing" in Table 2 is listed as either km or degree, and it should be unified for better comparison.

*The grid-cell spacings previously given at native model resolution in degrees, were converted to approximate distance in km.*

2. The information in Table 4 is not well summarized, and the differences among the six models should be known with only reading the table.

*We further split the relevant information of the differences among the cases in the attempt of making them more readable.*

3. In Figure 5, the colormaps for small and large count numbers are close, and clearer colormap is suggested.

*We reduced the number of colorscale bins in order to make the figure more readable.*

4. It seems that Europe and N. America do not show too many differences on the conclusions related to the mixing state and absorption simulations. There are much less data for evaluating the N. America case, which makes the discussion less solid (e.g. Figs 5b and 8b). Why not just focus on Europe, because it will not change the conclusions of the manuscript but makes it much easier for discussion.

*This intercomparison exercise is based on the collaboration of communities from both continents, thus, even if not central to the main scope of the manuscript, we prefer to keep it as it is. We believe that it is still a useful term of comparison for the reader and it thus not disturb much the presentation of the results. We added a short last paragraph to the result section 3: "Summarizing the comparison between the two continents, the selected AERONET observations generally show more absorbing (mean $\omega_{0,440}$ of 0.82 vs. 0.91) and spectrally dependent (mean $AAE_{675}^{440}$ of 1.19 vs. 1.10) aerosol over North America than Europe. The models broadly capture this variability, but display generally a larger bias over North America. The changes induced in the calculated optical quantities by the modifications tested here on the mixing state assumptions are consistent on the two regions.". The same concepts are reiterated also in the conclusions.*

5. For Figures 6 and 9, are the values in the y-axis in unit of percentage? For example, does 1.0 mean 100% or 1%?

*The legend on the y-axis was wrong. 1 means 100%, not 1%. We corrected the Figures, removing the "(%)" near "NMB" on the y-axis.*

---

## Author Comment (AC2) · 30 Oct 2018

Reviewer #1

Color legend:

- Reviewer's comments in black
- *Authors' responses in red italic*

Manuscript "Modelling black carbon absorption of solar radiation: combining external and internal mixing assumptions" by G. Curci et al.

General comments:

This work presents an interesting modelling study on the impact of mixing assumptions in the optical properties and radiative impact of aerosols in the atmosphere. The focus is on the absorption of Black Carbon (BC) and the impact of assuming external or internal mixtures. The authors use results from the regional models contributing to the third phase of the Air Quality Model Evaluation International Initiative (AQMEII), and through the use of an off-line tool to compute absorption properties (FlexAOD), model results are compared with AERONET sunphotometer retrievals.

*We would like to thank the reviewer for the insightful comments delivered on our work. We believe that addressing them made the results more robust and general with respect to the initial submission. We performed a number of additional sensitivity tests and comparison to measurements in order to clarify the detailed points, as illustrated below.*

The weaknesses of the study are:

(1) the inconsistency of using prescribed microphysical and optical properties for the aerosols different to the ones used by the models. The mass simulated by the set of models from AQMEII are strongly dependent on the microphysical properties of the aerosols assumed by each model (i.e., density, size distribution). The authors should justify that the harmonization applied in the microphysical properties of the aerosols has a second order impact on the understanding of the uncertainty associated with the mixing states assumed in coated BC particles.

*We believe that using the same assumptions, both in terms of physical-chemical properties and size distributions, for all models it is actually a strong point of the study. As illustrated in section 2.3, it happens very often that different models use different assumptions for aerosol optical properties calculations, which introduces a further element of ambiguity in the intercomparison. On the other hand, there is certainly a loss of detail on each model capability, since we only use the aerosol species bulk mass as input, in place of the (eventually) explicitly simulated size distribution. However, the latter information was not even stored and delivered to the intercomparison database. In order to avoid too much dependence on the assumed size distributions, we filtered out scenes having a large bias with respect to AERONET retrievals of effective radius and volume concentration, but we agree with the reviewer that this might still not be sufficient to convince about the robustness of our conclusions.*

*We thus expanded the work with a new section (3.1) reporting the results from further sensitivity tests, applying perturbed size distribution parameters to one selected model and looking at changes in main conclusions regarding the single scattering albedo and its spectral variation (AAE).*

*Regarding in particular the assumed size distribution, we report here the paragraph added to the new section 3.1:*

*"The first of this tests (GC), uses a completely different set of size distribution parameters. In particular, we substituted the log-normal parameters of **Errore. L'origine riferimento non è stata trovata.** with those used in the GEOS-Chem global chemistry transport model ([http://wiki.seas.harvard.edu/geos-chem/index.php/Aerosol_optical_properties](http://wiki.seas.harvard.edu/geos-chem/index.php/Aerosol_optical_properties)), as listed in **Errore. L'origine riferimento non è stata trovata.**. The result is a very little change in terms of absorption quantities, confirming that the results shown above are not very sensitive to the details of the assumed size distributions, in particular those regarding the material assumed to be in the shell.*

*In the second test devoted to size distributions (BC05), we modified only the size of BC. As shown in **Errore. L'origine riferimento non è stata trovata.**, the mean radius of the BC size distribution is assumed to be 0.0118 µm, which is comparable to the size of a single spherule (monomer) of BC. As mentioned in section 2.3, the real atmosphere observed form of BC goes from fractal aggregates of monomers to more compact forms as it ages. We thus repeated the calculations with an increased mean radius of 0.5 µm, in the middle of the range of radiuses explored by Li et al. (2016). The effect in the external mixing case is a slight increase of the $\omega_{0,440}$ and increased variability of the $AAE_{675}^{440}$. In the core-shell case, both $\omega_{0,440}$ and $AAE_{675}^{440}$ decrease, implying that larger BC cores increase the absorption and flatten its spectral dependence toward values more comparable with those deduced from AERONET measurements. As a caveat, the increase in the mean BC radius is what explains the difference between the CSBC and the CSBCV cases illustrated above. However, the $E_{abs}$ also increases by about 50% (not shown), thus a better simulation of $AAE_{675}^{440}$ is only apparently happening for the right reason, but this is certainly a point that should be further explored in future studies."*

(2) A second point that should clarified in the manuscript is the treatment of the Brown Carbon (BrC). If mineral dust is excluded, BrC is the second most absorbing aerosol in the atmosphere, and the treatment in models should be specified. Some organic aerosols have absorbing properties depending on the emission source, others may experience a browning effect through its aging. Some specific discussion on how models deal with that is required to understand the distinction done between bright organic aerosol and BrC. This has a significant impact on the results and conclusions of the work.

*We agree that the description of the treatment of BrC is not emphasized in the manuscript. Given the many uncertainties in the absorption properties of organic matter (OM), we preferred to focus the discussion on BC absorption. Regarding OM we made one extreme choice, i.e. we treat primary OM as BrC and secondary OM as non-absorbing. From model output delivered for the intercomparison, we only have POM and SOM (when simulated) total mass, without any tracking of the sources or the aging. We acknowledge, however, that the extent to which this assumption influences the main conclusions is not clear and we assessed it with further sensitivity tests on assumption on the treatment of OM absorption properties.*

*The outcome of additional tests is summarized in these paragraphs of the new section 3.1:*

*"The final subset of tests 7-9 are devoted at exploring the role of assumptions made on the absorption properties of BrC. In the baseline sensitivity tests presented above, we adopted the extreme choice of assigning BrC characteristics to the primary organic fraction. However, also the primary fraction is generally a mix of white and brown aerosol (e.g. Laskins et al., 2015). In test BRC0, we switch off the absorption due*

*to BrC, setting the imaginary part of primary OC to the low value of 10⁻⁸. The effect is a decreased absorption, denoted by the increase of $\omega_{0,440}$. More remarkably, there is a complete suppression of the spectral dependence of the absorption, denoted by the flattening of the simulated $AAE_{675}^{440}$ values. In the case of external mixing, $AAE_{675}^{440} \sim 1$, with very little variability, which is consistent with the presence of only externally mixed BC as an absorber (Liu et al., 2017b, Liu and Mishchenko, 2018). In the case of core-shell, most of the variability is also suppressed, but the mean value of $AAE_{675}^{440}$ is around 1.4, denoting the absorption amplification $E_{abs}$ by the shell around BC. According to recent calculations reported by Luo et al. (2018), the core-shell model is expected to exaggerate this amplification especially at shorter wavelengths, thus artificially increasing the calculated $AAE_{675}^{440}$.*

*In tests 8 (BRCS), we swapped the role of primary and secondary organic carbon as radiation absorber. The results are generally similar to the reference case, but there is an increased variability in the simulated values, reflecting the secondary nature of the aerosol, which is photochemically produced down-wind of the sources, and thus generally more variable. In the last test 9 (BRCSH), we further suppressed the hygroscopic growth assumed for the secondary organic fraction, while the primary was assumed hydrophobic in all the tests. The absence of water uptake by the aerosol increases the absorption (indeed water has a refractive index of 1.32-1.35 in the visible and it does not absorb light significantly), but does not affect much the its spectral variation."*

(3) The underestimation of the aerosol mass concentration at surface level and in the column observed in the AQMEII models introduces an important uncertainty on the results discussed in the manuscript. A lack of a comprehensive evaluation of the chemical composition of the aerosols is lying in the fundaments of the work. Thus, some discussion on that should be introduced in the revised manuscript and make it clear in the conclusions.

*As mentioned in the response to point 1, we tried to limit the influence of model bias in terms of mass and size distribution filtering out scenes with a large bias in terms of effective radius and volume concentration in the final analysis. However, the suggestion of the reviewer may help to present the results in a more clear way, and we thus compiled the following table with correspondences of AERONET stations and monitoring stations providing aerosol speciation measurements in Europe and North America:*

| AERONET Site | Latitude | Longitude | Database | Site | Latitude | Longitude |
|---|---|---|---|---|---|---|
| *Europe (20 sites)* | | | | | | |
| Andenes | 69.28 | 16.01 | - | | | |
| Barcelona | 41.39 | 2.12 | EMEP | Montseny | 41.77 | 2.35 |
| Brussels | 50.78 | 4.35 | - | | | |
| Brujassot | 39.51 | -0.42 | - | | | |
| Ersa | 43.00 | 9.36 | - | | | |
| Huelva | 37.02 | -6.57 | - | | | |
| Karlsruhe | 49.09 | 8.43 | - | | | |
| Kyiv | 50.36 | 30.50 | - | | | |
| Lecce University | 40.36 | 18.11 | - | | | |
| Malaga | 36.72 | -4.48 | - | | | |
| Messina | 38.20 | 15.57 | - | | | |
| Moldova | 47.00 | 28.82 | EMEP | Leova II | 46.49 | 28.28 |
| Munich University | 48.15 | 11.57 | - | | | |
| OHP Observatoire | 43.94 | 5.71 | - | | | |
| Palencia | 41.99 | -4.52 | EMEP | Campisabalos | 41.28 | -3.14 |
| Salon de Provence | 43.61 | 5.12 | - | | | |
| Sevastopol | 44.62 | 33.52 | - | | | |
| Thessaloniki | 40.63 | 22.96 | - | | | |
| Toravere | 58.26 | 26.46 | - | | | |
| Toulon | 43.14 | 6.01 | - | | | |
| *North America (9 sites)* | | | | | | |

| Bozeman | 45.66 | -111.05 | - | | | |
| BSRN BAO Boulder | 40.05 | -105.01 | IMPROVE | Rocky Mountain NP | 40.28 | -105.55 |
| Chapais | 49.82 | -74.98 | - | | | |
| Easton Airport | 38.81 | -76.07 | IMPROVE | Washington DC | 38.88 | -77.03 |
| Egbert | 44.23 | -79.75 | IMPROVE | Egbert | 44.23 | -79.78 |
| El Segundo | 33.91 | -118.38 | IMPROVE | San Gabriel | 34.30 | -118.02 |
| Halifax | 44.64 | -63.59 | - | | | |
| Railroad Valley | 38.50 | -115.96 | IMPROVE | Great Basin NP | 39.01 | -114.22 |
| Saturn Island | 48.78 | -123.13 | IMPROVE | Olympic | 48.01 | -122.97 |

*For Europe, a reasonable correspondence is found only for 3 stations, while in North America for most stations (6 on 9). We collected the available observations for the year 2010 at these stations and carried out the comparison with concentrations of aerosol species at the first model level. The results are summarized in the supplementary revised Table S1 and the new Figure S7, and briefly discussed in section 2.2 in a new paragraph:*

*"Additional indications about models skills are gathered from the comparison with PM composition measurements available near the AERONET stations, for which we have stored the simulated PM speciation profiles of AQMEII models. The comparison is carried out at 3 stations over Europe and 5 stations over North America, and results summarized in Table S1 and Figure S7. Over North America, the two models have yearly average values mostly within ±1 μg m-3. Over Europe, most values are also within the same range, but there is a tendency toward overestimation of inorganic secondary species (sulfate, nitrate, ammonium) and black carbon, and underestimation of the organic carbonaceous fraction."*

*Given the very limited number of stations, we haven't emphasized the comparison to more than an additional indication on models skills.*

*We then used this, although partial, information to evaluate the impact of aerosol mass model bias on the main conclusions, as reported in a paragraph of the new section 3.1:*

*"We run the tests in the two extreme and more physically relevant mixing assumption adopted above, i.e. external mixing (EXT) and core-shell (CSBC). The first subset of tests is related to the influence of the model bias in terms of aerosol species mass. From Table S1, we estimate that model IT2 overestimates sulfate by a factor of 3, ammonium and BC by a factor of 2, while nitrate and organic fraction is in the range of observations. The tests 2-4 thus explore the effect of the mass adjustment on $\omega_{0,440}$ and $AAE_{675}^{440}$, as illustrated in the related scatterplot in **Errore. L'origine riferimento non è stata trovata.**. The correction of secondary inorganic aerosol mass yields a negligible change in terms of calculate absorption properties, while the correction of BC mass introduces more change: the reduction of BC mass, as might be expected, reduces the absorption ($\omega_{0,440}$ increases) and makes its spectral variation more steep ($AAE_{675}^{440}$ increases). The change is of the order of 3-4%, which is comparable to the magnitude of models' $\omega_{0,440}$ bias, but it is of the same sign and magnitude for external and core-shell mixing. The bias of BC mass is thus unlikely to alter the main conclusions regarding calculated absorption properties illustrate above."*

The paper is generally well-written. I recommend it to publish in ACP after the previous weakness/questions and following specific and technical comments are addressed.

Specific comments:

- Page 4 Line 9: What is the criterion followed to select the AERONET stations? Are only urban stations selected?

*We required a minimum percentage of data available for the year, which we set to 10%. The information was missing in the manuscript and we now added it in the revised version, after the same line:*

*"We select only those stations having a minimum of 10% of valid data in 2010."*

- Page 5 Line 4: Is not the 1.2 thershold for AAE too small for BrC? Lack and Cappa (2010) showed that AAE values, ranging from 1–1.6, can be observed for internally mixed OC/BC particles and suggested that aerosols with AAE exceeding 1.6 should be classified as BrC.

*Our aim was primarily to discard dust-dominated scenes, and leave the distinction between scenes dominated only by BC absorption and those influenced by both BC and BrC as a secondary element of interpretation of the results. This choice was based on the large uncertainty related to BrC modelling with respect to BC, with the aim of further analysing the effect of assumptions specifically in BC-only dominated scenes, without the complications implied by the interpretation of non-dust scenes taken altogether. We thus followed the suggestion formulated by Badhur et al. (2012) and set the AAE threshold to 1.2 to delimit the two cases. According to Lack and Cappa (2010) this threshold could be further lowered down to 1, which is a more conservative value. However, we haven't changed any separation threshold in the revised paper, because the discussion on the results is actually carried out considering "BC" and "BC+BrC" scenes altogether. The separation just remains as a qualitative guidance for the reader. Moreover, this comment is strictly related to the role played by BrC in the conclusions of the paper, which in turn was assessed through additional sensitivity tests on BrC optical assumptions as explained in response to main comment 2.*

- Page 5 Line 6: What is the difference between your approach and Wang et al. (2016)? Would this affect the results of the work?

*As stated, the difference is that here we do not attempt at segregating "BC" and "BrC" scenes, thus in the submitted version of the manuscript this is not affecting the results, because they are summarized collectively as "BC" and "BC+BrC" scenes altogether.*

- Page 5 Line 22: What are the chemical boundary conditions used in the models?

*As already stated at lines 22-23, the boundary conditions are taken from C-IFS global simulations (Flemming et al., 2015), specifically run for AQMEII-3 (Galmarini et al., 2017).*

- Page 6 Line 4: A justification of the different results over Europe of model ES1 and over US of model DK1 compared with the others would be useful. Differences are quite significant.

*Regarding the model ES1, the difference with other models is "due to a known overestimation of desert dust", while for DK1 the bias is "mostly attributable to missing secondary organic aerosol mass". The information is know added in the same sentence. Regarding DK1 over the US*

- Page 6 Line 8: There are observational networks in EU and USA that measure chemical composition of particles (i.e., EBAS, EMEP, IMPROVE). Why are not used here to quantify the errors on BC, organic mass,

sulfate and dust concentrations? The work presented in the manuscript would benefit from a clear quantificaction of the errors of the models on the chemical composition of the aerosols, at least at surface level.

*Please see response to main point 3 above.*

- Page 6 Line 12: Zhang et al. (2017) showed that some aircraft experimental data suggest that BrC constitute a significant part of absorbing carbonaceous aerosols, especially at high altitudes (> 5km). Some model disagreement with observations seems related to the treatment of BrC. The assumptions used in the present work related to BrC should be described in more detail.

*Please see response to main point 2 above.*

- Page 6 Line 15: It would be useful to understand the reasons of the differences of ES1 and TR1 models predicting the secondary inorganic species. If emissions are the same ones for all the models, the differences should not be so large among models.

*This is certainly a good point, but it was difficult to explore here with the information available. Even if the models share same anthropogenic emissions and boundary conditions, they may vary significantly in the details of implementation of chemical processes. Specifically for secondary inorganic species, what could make a large difference is the treatment of thermodynamic equilibrium and aqueous phase chemistry. Investigation of this kind of differences is however beyond the scope of this study, here we limit our attention to the fields delivered by the model as they are, in order to provide a diverse ensemble of aerosol scenes for optical properties calculations.*

- Page 6 Line 21: Here again, there is a need to clarify how BrC is treated in the models. There is some evidence that some BrC comes from secondary organic aerosols (Laskins et al., 2015).

*Please see response to main point 2 above.*

- Page 6 Line 31: It would be useful to include here the description of how BC absorption enhancement and BC core mass fraction are calculated. Right now, the definitions are in the caption of Figure 2.

*We added the information as suggested. "Here we calculate the BC absorption enhancement as the ratio of absorption optical depth calculated assuming internal mixing to the one calculated using external mixing:*

$$E_{abs} = \frac{\tau_{abs}(\lambda, internal\ mixing)}{\tau_{abs}(\lambda, external\ mixing)} \hspace{4cm} (6)$$

*The BC core mass fraction is defined for core-shell calculations as the ratio of BC mass (the core) to total aerosol mass (shell + core)."*

- Page 7 Line 7: It seems that EU models reproduce the AOD for the wrong reasons considering the significant bias in surface PM2.5. This points to the need of an evaluation of the chemical composition of this PM2.5. What are the implications to the results obtained from the work? It can be expected than a different aerosol vertical distribution will considerably modify the findings of the work.

*Please see response to main point 3 above.*

- Page 7 Line 27: Are the optical properties of the shell species computed as an homogeneous internal mixing?

*Yes. The adjective "homogeneously mixed" is added before "shell" in order to explicitly state this assumption.*

- Page 8 Line 4: From the results, it turns out that the methodology to combine size distributions is a critical point. This should be remarked in the conclusion of the manuscript as an open issue that deserves further analysis.

*We agree and we added a remark on this point at the end of the fourth paragraph of conclusions (page 12):*

*"The methodology adopted to combine several size-distributions in a homogeneously mixed shell is thus a point that deserves further analysis in the future."*

- Page 9 Line 18: Would the use of the original models size distribution reduce this bias instead of using a uniform assigned distributions? The assumption of a bulk aerosol implies a loss of detail from the original model results.

*Please see response to main point 1 above.*

- Page 9 Line 26: It would be interesting to present the statistics of the models for this final subset of data. It is expected that the bias on PM2.5 and AOD will be reduced and the confidence with the findings may be stronger.

*The statistics for SSA and AAE are already presented in Tables 6 and 7. The analysis was enriched using available ground-based observations, as outlined in the response to main point 3 above.*

- Page 11 Line 5: The treatment of BrC in the simulations is not clear. From Table 4, the use of an imaginary refractive index of 0.021 following Highwood et al. (2009) for POM is already quite absorbing. Tang et al. (2016) and Kirchstetter et al. (2004) suggest an imaginary value of 0.03 for BrC from biomass burning. It seems that the assumptions selected for the optical properties of POM contributes to a more absorbing aerosol than the other way around.

*As mentioned in response to main comment 2, the way we treat BrC here is assigning non-negligible imaginary refractive index to POM, as suggested in the review by Highwood (2009). In the intercomparison, only primary and secondary OC were tracked, thus we don't have explicit simulation of the contribution from biomass burning or other specific sources, and we don't have an explicit treatment of aging. The influence of this assumption on the main was discussed in response to main point 2.*

In Table 4, it seems there is a typo in the imaginary refractive index of POM, which is set to 0.21. Highwood et al. (2009) suggest a value of 0.021 at 550 nm for the organic aerosol.

*Thanks much for spotting this, it was a typo. Now corrected to 0.021.*

\- Page 11 Line 14: Results of combining external and internal mixtures (PIM cases) perform quite similar to CSBC. It would be worth mention it here.

*The fact that PIM cases show features similar to CSBC, as compared to HOM and CSBCV, is due to its calculation: as illustrated in Table 4, PIM cases are the composition of the external mixing case with the CSBC internal mixing case. We thus prefer not to overemphasize this point here.*

\- Page 11 Line 15: A general discussion comparing the EU and US results would be interesting as closing paragraph.

*We added a short last paragraph to the result section 3: "Summarizing the comparison between the two continents, the selected AERONET observations generally show more absorbing (mean $\omega_{0,440}$ of 0.82 vs. 0.91) and spectrally dependent (mean $AAE_{675}^{440}$ of 1.19 vs. 1.10) aerosol over North America than Europe. The models broadly capture this variability, but display generally a larger bias over North America. The changes induced in the calculated optical quantities by the modifications tested here on the mixing state assumptions are consistent on the two regions."*

\- Page 11 Line 26: This last sentence should be included in the abstract to clarify that the models are compared under BC dominant conditions.

*We added the sentence "discarding from the analysis scenes dominated by dust" at the end of the first paragraph of the abstract.*

\- Page 11 Line 30: What are the recommendations for the modellers to improve the absorption of they systems? There is still significant variability among models, but some insights from the results indicate internal coating or mixtures of external and internal coating are recommended approaches. This can be highlighted in the conclusions. A comment on the impact seen between CSBC and CSBCV is an important result of the work. Only the method used to derive a resulting distribution of an internal mixture is still a significant source of uncertainty.

*The first point is remarked in the last paragraph of the conclusions "In conclusion, this work suggests that the combination of external and core-shell mixing state have the potential for a realistic representation of atmospheric aerosol absorption and its spectral dependence. However, the validation of model calculations using only sunphotometers retrievals as point of comparison is not exhaustive.". Regarding the second point, we added the following last remark "Moreover, the use of explicitly simulated aerosol size distributions should be included in future work, as opposed to the use of assigned size distributions as done here, in order to further investigate the effect of core mass fraction changing with aerosol size. The introduction of more detailed treatment of the aging structure of BC and BrC is also recommended, in combination with algorithms more accurate than the core-shell model, such as the multiple-sphere T-matrix method."*

\- Page 23 Table 3: Are the AOD values presented here computed with FlexAOD? Why are there such differences compared with Table S1, where US3 reports an AOD at 440 nm of 0.05?

*The values given in Table 3 are all from internal calculations of each model, not carried out using FlexAOD. As specified in section 2.3, the scope of this Table was exactly to point out the potential inconsistencies of aerosol mass and optical depth biases when intercomparing the models, because of the different underlying assumptions. This explains the differences with Table S1. However, since we believe that the presence of Table 3 might be confusing to the reader, we moved it to the online supplement, and it is now called Table S1, and referenced as such in the manuscript.*

- Page 23 Table 4: There is a typo in the imaginary part of the refractive index of POM. The refractive index of POM is more representative of a pure BrC rather than a mixture of white organic aerosol and BrC. Nakayama et al. (2012) or Liu et al. (2013) suggest a refractive index of 1.486 - i 2.5e-5 at 550 nm for secondary organic aerosols. Concerning the growth factor, is only the growth factor at 90% considered? The growth factor starts to be non-negligible at 75% relative humidity. Some description on role of the growth factor on the resulting refractive index of an internal mixture would be relevant for the work. Why are BC and POM considered hydrophobic?

*As stated in the comment above, we confirm the POM imaginary part of the refractive index was wrong (0.21 instead of 0.021). The hygroscopic growth factor at 90% RH is given in the table for illustration. In the model RH is allowed to continuously vary and growth factors are interpolated from available tables, which are taken from Hess et al. (1998). The information about the source of hygroscopic growth factors is now emphasized on page 7: "(source of data are Highwood (2009) and Hess et al., (1998), the latter for hygroscopic growth factors)". Moreover, the effect of the assumption on hygroscopic growth was tested in an additional sensitivity tests, as explained in the response to main point 2.*

- Page 24 Table 6: Some clarification on the numerical failures of the optical calculations is needed.

*We added the following sentence at the end of the first paragraph on page 8 (FlexAOD description section): "For some extreme situations, such as very small or zero core size, the code do not attempt to perform extrapolations and returns a failed calculation. Depending on the combination of aerosol species, the number of valid calculation may thus slightly vary (see Tables 5 and 6)."*

Technical corrections:

- Page 2 Line 15: I suggest to move this last sentence to the last paragraph of the introduction.

*The first paragraph of the introduction is intended as a very brief summary and outline of the introduction itself, and of the broad motivation and aim of the paper. The concepts given here are extensively expanded in the subsequent paragraphs, we thus prefer to leave this sentence in its current place. To clarify the logic, we reiterated the reference to Fierce et al. (2017) in first sentence of the third paragraph, where this message is expanded in details.*

- Page 3 Line 22: Replace "external or internal" by "external and internal".

*Corrected.*

- Page 5 Line 23: Delete the second point after the reference (Flemming et al., 2015).

*Done.*

- Page 7 Line 16: Check syntax of the references "(source of data Highwood (2009) and Hess et al. (1998))".

*We modified the sentence as "Table 4 (source of data are Highwood (2009) and Hess et al., (1998))"*

- Page 7 Line 29: Delete the point after the word "mixing".

*Done.*

- Page 8 Line 11: Finalize the sentence with "and particle volume are:".

*We added "are calculated as"*

- Page 9 Line 15: Include a closing point at the end of the paragraph.

*Done.*

- Page 22 Table 2: Complete the table with the "Aerosol model description" for the University of Aarhus WRF-DEHM model.

*The missing information was added to the table.*

- Figure 2, 3, 5 and 8: The quality or resolution of the figures is low for final publication.

*Probably the quality of the figures was degraded when inserted into Word. For final production the original files will be provided.*

References:

Kirchstetter, T. W., Novakov, T., and Hobbs, P. V. Evidence that the spectral dependence of light absorption by aerosols is affected by organic carbon. Journal of Geophysical Research, 109 (2004).

Lack, D. A.; Cappa, C. D. Atmos. Chem. Phys. 2010, 10, 4207.

Laskin, A., Laskin, J. & Nizkorodov, S. A. Chemistry of Atmospheric Brown Carbon. Chem. Rev. 115, 4335–4382 (2015).

Liu, P., Zhang, Y., and Martin, S. T. Complex Refractive Indices of Thin Films of Secondary Organic Materials by Spectroscopic Ellipsometry from 220 to 1200 nm. Environmental Science & Technology, 47, 13594–13601 (2013).

Nakayama, T., Sato, K., Matsumi, Y., Imamura, T., Yamazaki, A., and Uchiyama, A. Wavelength Dependence of Refractive Index of Secondary Organic Aerosols Generated during the Ozonolysis and Photooxidation of alpha-Pinene. Scientific Online Letters on the Atmosphere, 8, 119–123, (2012).

Tang, M., Alexander, J. M., Kwon, D., Estillore, A. D., Laskina, O., Young, M. A., Kleiber, P. D., and Grassian, V. H. Optical and Physicochemical Properties of Brown Carbon Aerosol: Light Scattering, FTIR Extinction Spectroscopy, and Hygroscopic Growth. The Journal of Physical Chemistry A, 120, 4155–4166, (2016).

Zhang, Y. et al. Top-of-atmosphere radiative forcing affected by brown carbon in the upper troposphere. Nat. Geosci. 10, 486–489 (2017).

---

## Author Comment (AC3) · 30 Oct 2018

Comment

This manuscript is generally well written, but there seems to be one important piece missing, which is related to BC particle structures. This study assumed spheres for externally mixed BC and core-shell coating structure for internally mixed BC. However, recent observations (e.g., China et al., 2015; Wang et al., 2017) have shown irregular fractal aggregates for externally mixed BC and various non-core-shell coating structures for internally mixed BC. Further modeling studies (e.g., Scarnato et al., 2013; He et al., 2015, 2016) have indicated a large variation in BC optical properties due to the observed complex fractal and coating structures. Thus, the assumptions in the current manuscript may lead to uncertainty in the calculations of BC optical properties. Thus, I suggest that the authors include these recent studies and add some discussions on this important issue.

*We thank Dr. He for this useful comment on our manuscript. We acknowledge that the point raised is relevant and not properly illustrated in our first version of the paper. In section 2.3, where we explain details on the calculations carried out, we added a paragraph on the issue:*

*"Specifically regarding the modelling of BC shape and mixing state, here we adopt the simplified approach widely used in regional and global models of assuming spherical particles and centred core-shell arrangement for internal mixing calculations, which makes the computation fast enough for 3-D applications in year-long simulations. However, observations show that BC in the real atmosphere displays a wide variety of shapes: freshly emitted hydrophobic fractal aggregates, consisting of hundreds of spherules having diameters of a few tens of nm (e.g. Posfai et al., 2003, Adachi and Buseck, 2013), typically evolve in the atmosphere assuming more compact structures, and internally or semi-internally coating with hydrophilic material (e.g. Adachi et al., 2010, China et al., 2015, Wang et al., 2017). These transformations affect the variability of the absorption properties of BC, as illustrated in several numerical studies that include detailed description of the shapes and mixing state of BC and that use advanced algorithms, such as the multiple-sphere T-matrix (MSTM) and the discrete dipole approximation (DDA), to compute the optical properties (Scarnato et al., 2013, He et al., 2015, He et al., 2016, Li et al., 2016, Kahnert, 2017, Liu et al., 2017b, Liu et al., 2018, Liu and Mishchenko, 2018). Moreover, also the shapes of BrC may vary in the real atmosphere, but their classification and investigation of numerical aspects in the calculation of optical properties is still at its beginning (Laskin et al., 2015, Liu and Mishchenko. 2018)."*

*Moreover, this comment, together with related comments from the anonymous reviewers, suggested a new sensitivity tests on assumptions specifically related to BC size distribution, as illustrated in a paragraph of the new section 3.1:*

*"In the second test devoted to size distributions (BC05), we modified only the size of BC. As shown in **Errore. L'origine riferimento non è stata trovata.**, the mean radius of the BC size distribution is assumed to be 0.0118 μm, which is comparable to the size of a single spherule (monomer) of BC. As mentioned in section 2.3, the real atmosphere observed form of BC goes from fractal aggregates of monomers to more compact forms as it ages. We thus repeated the calculations with an increased mean radius of 0.5 μm, in the middle of the range of radiuses explored by Li et al. (2016). The effect in the external mixing case is a slight increase of the $\omega_{0,440}$ and increased variability of the $AAE_{675}^{440}$. In the core-shell case, both $\omega_{0,440}$ and $AAE_{675}^{440}$ decrease, implying that larger BC cores increase the absorption and flatten its spectral dependence toward values more comparable with those deduced from AERONET measurements. As a caveat, the increase in the mean BC radius is what explains the difference between the CSBC and the CSBCV cases illustrated above. However, the $E_{abs}$ also increases by about 50% (not shown), thus a better simulation of $AAE_{675}^{440}$ is only apparently happening for the right reason, but this is certainly a point that should be further explored in future studies."*

References

China, S., et al.: Morphology and mixing state of aged soot particles at a remote marine free troposphere site: Implications for optical properties, Geophys. Res. Lett., 42, 1243–1250, doi:10.1002/2014gl062404, 2015.

He, C., et al.: Variation of the radiative properties during black carbon aging: theoretical and experimental intercomparison, Atmos. Chem. Phys., 15, 11967-11980, doi:10.5194/acp-15-11967-2015, 2015.

He, C., et al.: Intercomparison of the GOS approach, superposition T-matrix method, and laboratory measurements for black carbon optical properties during aging, J. Quant. Spectrosc. Radiat. Transf., 184, 287–296, doi:10.1016/j.jqsrt.2016.08.004, 2016.

Scarnato, B. V., et al.: Effects of internal mixing and aggregate morphology on optical properties of black carbon using a discrete dipole approximation model, Atmos. Chem. Phys., 13, 5089–5101, doi:10.5194/acp-13-5089-2013, 2013.

Wang, Y., et al.: Fractal dimensions and mixing structures of soot particles during atmospheric processing, Environ. Sci. Technol. Lett., 4, 487-493, doi:10.1021/acs.estlett.7b00418, 2017.

---

## Author Response (AR2)

Cover letter to the Editor

Dear Editor,

Thanks for appreciating the revision effort we made aimed at improving the clarity and robustness of the presented results. We submitted a final revision of the manuscript, integrating the final reviewers' comments. Here below we include the description of the further changes to the manuscript in response to these comments.

With kind regards,

Gabriele Curci on behalf of all co-authors

Response to reviewer's final recommendations

Color legend:

- Reviewer's comments in black
- *Authors' responses in red italic*

Report #1

The authors have addressed all of my suggestions. There is only one minor suggestion that is not well answered.

Yes, I agree that analysis of the AERONET error is beyond the scope of this study, and the authors explained that in the response. However, I think the retrieval errors may be well studied by other studies, and the authors may just slightly discusses the uncertainties in the manuscript based on literature review. This will certainly help to better understand the comparison, because the differences between this study and AERONET are not merely from the simulations. Again, as I mentioned, this is minor, and the authors can decide whether add or not on their own.

*In the manuscript we previously reported the nominal uncertainty associated to AERONET single scattering albedo measurements, which is the relevant quantity used in this work, on page 4, lines 18-20:*

*"The uncertainty associated with the single scattering albedo is estimated to increase from ±0.03 for $\tau(\lambda=440\ nm) \geq 0.5$ to ±0.05-0.07 for $\tau(\lambda=440\ nm) \leq 0.2$ (Dubovik et al., 2000)."*

*Since most of the observations over Europe and North America are in the lower range of aerosol optical depths, the associated uncertainty for the single scene is higher (±0.05-0.07). In a statistical sense, on averaged values this uncertainty should be decreased by a factor of $\sqrt{n}$, where n is the number of measurements. The average changes in single scattering albedo calculated in the sensitivity tests with respect to the mixing state assumption (Table 5) are mostly much larger than those values, i.e. they should be safely beyond the observation uncertainty. We thus prefer not to overemphasize this point more than already done in the current version of the manuscript.*

Report #2

After revising the authors' responses to the two referees and the revised manuscript, I suggest to accept the manuscript for final publication in Atmospheric Chemistry and Physics after addressing the following technical comments:

- Captions of Table 4 and 7 shall be extended with the description of the contents of the tables as done in the rest of figures and tables. Table captions should be self-descriptive.

*We expanded the captions clarifying their contents and making reference to the Figures where the related results are illustrated:*

*"Table 4. List of baseline sensitivity simulations on aerosol optical properties calculations. The case with full external mixing (EXT) is taken as reference, the other cases are sensitivity tests in which we changed one assumption per case related to the aerosol mixing state. The difference between CSBC and CSBCV cases is further illustrated in Figure 4. Results are shown in Figure 5-Figure 10."*

*"Table 7. List of additional sensitivity tests on BrC and size distribution assumptions. Here the changes are evaluated with respect to both the EXT and CSBC cases described in Table 4, changing one assumption per case. Results are shown in Figure 11."*

- I also suggest a final proofreading of the manuscript to correct some mistyping and spelling errors still present in the revised manuscript.

*We did a final proofreading of the manuscript and corrected typos and references.*